# Going beyond persistent homology
# using persistent homology

**Johanna Immonen**[*]
University of Helsinki
johanna.x.immonen@helsinki.fi

**Amauri H. Souza**[*]
Aalto University
Federal Institute of Ceará
amauri.souza@aalto.fi

**Vikas Garg**
Aalto University
YaiYai Ltd
vgarg@csail.mit.edu

## Abstract

Representational limits of message-passing graph neural networks (MP-GNNs), e.g., in terms of the Weisfeiler-Leman (WL) test for isomorphism, are well understood. Augmenting these graph models with topological features via persistent homology (PH) has gained prominence, but identifying the class of attributed graphs that PH can recognize remains open. We introduce a novel concept of color-separating sets to provide a complete resolution to this important problem. Specifically, we establish the necessary and sufficient conditions for distinguishing graphs based on the persistence of their connected components, obtained from filter functions on vertex and edge colors. Our constructions expose the limits of vertex- and edge-level PH, proving that neither category subsumes the other. Leveraging these theoretical insights, we propose RePHINE for learning topological features on graphs. RePHINE efficiently combines vertex- and edge-level PH, achieving a scheme that is provably more powerful than both. Integrating RePHINE into MP-GNNs boosts their expressive power, resulting in gains over standard PH on several benchmarks for graph classification.

## 1 Introduction

Topological data analysis (TDA) is a rapidly growing field that provides tools from algebraic topology for uncovering the *shape* (or structure) of data, allowing for efficient feature extraction. Its flagship tool is persistent homology (PH) [8], which seeks to characterize topological invariants (e.g., connected components, loops) of an underlying manifold based on data samples. Notably, PH has been successfully applied in many scientific domains, including computer vision [17, 27], drug design [23], fluid dynamics [24], and material science [25].

For graphs, PH has been used to provide global topological signatures for graph-level prediction tasks [2, 12, 14, 33, 39] and act as local message modulators in graph neural networks (GNNs) for node-level tasks [4, 40]. By leveraging learnable filtration/vectorization maps, PH has also been integrated into neural networks as a building block in the end-to-end learning process [2, 3, 13, 15, 26]. These strategies allow us to exploit topological features to boost the predictive capabilities of graph models. However, in stark contrast with the developments on the representational power of GNNs [1, 11, 28–30, 34, 35, 37], the theoretical properties of PH on graphs remain much less explored. For instance, open questions include: Which graph properties can PH capture? What is the characterization of pairs of graphs that PH cannot separate? Can we improve the expressivity of PH on graphs?

In a recent work, Rieck [32] discusses the expressivity of PH on graphs in terms of the Weisfeiler-Leman (WL) hierarchy [36]. The paper shows that, given different k-WL colorings, there exists a filtration such that the corresponding persistence diagrams differ. This result provides a lower bound for the expressivity in terms of WL hierarchy, but it does not describe the class of graphs which can be distinguished via PH. In this paper, we aim to fully characterize this class of graphs.

---

[*]Equal contribution.

37th Conference on Neural Information Processing Systems (NeurIPS 2023).

| Theoretical contributions of this work | |
|---|---|
| **On vertex-level filtrations (Section 2 and Section 3.1):** | |
| Inconsistency issues due to injective vertex filtrations | Lemma 1 |
| Real holes ($d = \infty$) $\cong$ Component-wise colors | Lemma 2 |
| Almost holes ($b \neq d, d \neq \infty$) $\cong$ Separating sets | Lemma 3 |
| Distinct almost holes $\Rightarrow$ Color-separating set | Lemma 4 |
| Birth time of persistence tuples $\cong$ Vertex color | Lemma 5 |
| The expressive power of vertex-color filtrations | Theorem 1 |
| **On edge-level filtrations (Section 3.2):** | |
| Almost holes $\cong$ Disconnecting sets | Lemma 6 |
| Reconstruction of disconnecting sets | Lemma 7 |
| The expressive power of edge-color filtrations | Theorem 2 |
| **Vertex-level vs. edge-level filtrations (Section 3.3):** | |
| Vertex-level persistence $\not\succ$ edge-level persistence, and vice-versa | Theorem 3 |
| **New method (RePHINE) (Section 4):** | |
| RePHINE is isomorphism invariant | Theorem 4 |
| RePHINE $\succ$ vertex-, edge-, and vertex- $\cup$ edge-level diagrams | Theorem 5 |

Figure 1: Overview of our theoretical results.

We study the expressive power of PH on attributed (or colored) graphs, viewed as 1-dim simplicial complexes. We focus on the class of graph filtrations induced by functions on these colors. Importantly, such a class is rather general and reflects choices of popular methods (e.g., topological GNNs [15]). We first analyze the persistence of connected components obtained from vertex colors. Then, we extend our analysis to graphs with edge colors. To obtain upper bounds on the expressive power of color-based PH, we leverage the notion of separating/disconnecting sets. This allows us to establish the necessary and sufficient conditions for the distinguishability of two graphs from 0-dim persistence diagrams (topological descriptors). We also provide constructions that highlight the limits of vertex- and edge-color PH, proving that neither category subsumes the other.

Based on our insights, we present RePHINE (short for "**Re**fining **PH** by **I**ncorporating **N**ode-color into **E**dge-based filtration"), a simple method that exploits a subtle interplay between vertex- and edge-level persistence information to improve the expressivity of color-based PH. Importantly, RePHINE can be easily integrated into GNN layers and incurs no computational burden to the standard approach. Experiments support our theoretical analysis and show the effectiveness of RePHINE on three synthetic and six real datasets. We also show that RePHINE can be flexibly adopted in different architectures and outperforms PersLay [2] — a popular topological neural net.

In sum, **our contributions** are three-fold:

**(Theory)** We establish a series of theoretical results that characterize PH on graphs, including bounds on the expressivity of vertex- and edge-level approaches, the relationship between these approaches, and impossibility results for color-based PH — as summarized in Figure 1.

**(Methodology)** We introduce a new topological descriptor (RePHINE) that is provably more expressive than standard 0-dim and 1-dim persistence diagrams.

**(Experiments)** We show that the improved expressivity of our approach also translates into gains in real-world graph classification problems.

## 2 Preliminaries

We consider arbitrary graphs $G = (V, E, c, X)$ with vertices $V = \{1, 2, \ldots, n\}$, edges $E \subseteq V \times V$ and a vertex-coloring function $c : V \to X$, where $X$ denotes a set of $m$ colors or features $\{x_1, x_2, \ldots, x_m\}$ such that each color $x_i \in \mathbb{R}^d$. We say two graphs $G = (V, E, c, X)$ and $G' =$

$(V', E', c', X')$ are isomorphic (denoted by $G \cong G'$) if there is a bijection $g : V \to V'$ such that $(v, w) \in E$ iff $(g(v), g(w)) \in E'$ and $c = c' \circ g$. Here, we also analyze settings where graphs have an edge-coloring function $l$.

A *filtration* of a graph $G$ is a finite nested sequence of subgraphs of $G$, that is, $G_1 \subseteq G_2 \subseteq ... \subseteq G$. Although the design of filtrations can be flexible [12], a typical choice consists of leveraging a vertex filter (or filtration) function $f : V \to \mathbb{R}$ for which we can obtain a permutation $\pi$ of $n$ vertices such that $f(\pi(1)) \leq f(\pi(2)) \cdots \leq f(\pi(n))$. Then, a filtration induced by $f$ is an indexed set $\{G_{f(\pi(i))}\}_{i=1}^n$ such that $G_{f(\pi(i))} \subseteq G$ is the subgraph induced by the set of vertices $V_{f(\pi(i))} = \{v \in V \mid f(v) \leq f(\pi(i))\}$. Note that filtration functions which give the same permutation of vertices induce the same filtration. Persistent homology keeps track of the topological features of each subgraph in a filtration. For graphs $G$, these features are either the number of connected components or independent cycles (i.e., 0- and 1-dim topological features, denoted respectively by the Betti numbers $\beta_G^0$ and $\beta_G^1$) and can be efficiently computed using computational homology. In particular, if a topological feature first appears in $G_{f(\pi(i))}$ and disappears in $G_{f(\pi(j))}$, then we encode its persistence as a pair $(f(\pi(i)), f(\pi(j)))$; if a feature does not disappear in $G_{f(\pi(n))} = G$, then its persistence is $(f(\pi(i)), \infty)$. The collection of all pairs forms a multiset that we call *persistence diagram* [5]. We use $\mathcal{D}^0$ and $\mathcal{D}^1$ to refer to the persistence diagrams for 0- and 1-dim topological features respectively. Appendix A provides a more detailed treatment for persistent homology.

Recent works have highlighted the importance of adopting injective vertex filter functions. Hofer et al. [13] show that injectivity of parameterized functions $f_\theta : V \to \mathbb{R}$ is a condition for obtaining well-defined gradients with respect to the parameters $\theta$, enabling end-to-end filtration learning. Also, Horn et al. [15] show that for any non-injective function, we can find an arbitrarily close injective one that is at least as powerful at distinguishing non-isomorphic graphs as the original (non-injective) function. However, Lemma 1 shows that filtrations induced by injective functions on vertices may result in inconsistent persistence diagrams; namely, different diagrams for isomorphic graphs.

**Lemma 1 (Injective vertex-based filtrations can generate inconsistent persistence diagrams).** *Consider persistence diagrams obtained from injective vertex filter functions. There are isomorphic graphs $G \cong G'$ such that their persistence diagrams are different, i.e., $\mathcal{D}_G \neq \mathcal{D}_{G'}$.*

To avoid inconsistent diagrams, we need to employ permutation equivariant filter functions — see [32, Lemma 2]. Common choices include vertex degree [12], eigenvalues of the graph Laplacian [2], and GNN layers [13], which are permutation equivariant by construction. Another option is to define graph filtrations based on vertex/edge colors [15], which are also equivariant by design, i.e., if $G \cong G'$ with associated bijection $g$, then $c(v) = c'(g(v)) \, \forall v \in V$. Notably, color-based filtrations generalize the GNN-layers case since we could redefine vertex/edge-coloring functions to take the graph structure as an additional input. Thus, we now turn our attention to color-based filtrations.

**Color-based filtrations.** Let $f : X \to \mathbb{R}$ be an injective function. Therefore, $f$ must assign a strict total order for colors, i.e., there is a permutation $\pi : \{1, \ldots, m\} \to \{1, \ldots, m\}$ such that $f(x_{\pi(1)}) < \cdots < f(x_{\pi(m)})$. We define the *vertex-color filtration* induced by $f$ as the indexed set $\{G_i\}_{i=1}^m$ where $G_i = (V_i, E_i, c_i, X_i)$, with $X_i = \{x_{\pi(1)}, x_{\pi(2)}, \ldots, x_{\pi(i)}\}$, $V_i = \{v \in V \mid c(v) \in X_i\}$, $E_i = \{(v, w) \in E \mid c(v) \in X_i, c(w) \in X_i\}$, and $c_i = \{(v, c(v)) \mid v \in V_i\}$. Similarly, we can define the *edge-color filtration* induced by $f$ as $\{G_i\}_{i=1}^m$ where $G_i = (V, E_i, l_i, X_i)$ with $X_i = \{x_{\pi(1)}, \ldots, x_{\pi(i)}\}$, $E_i = \{(v, w) \in E \mid l(v, w) \in X_i\}$, and $l_i = \{((v, w), l(v, w)) \mid (v, w) \in E_i\}$.

We denote the elements of a persistence diagram $\mathcal{D}$ as pairs $(f(x^{(b)}), f(x^{(d)}))$, where $x^{(b)}, x^{(d)} \in X$ are the colors associated with the birth and death of a hole (topological feature) in a filtration induced by $f(\cdot)$. In the following, we use the notation $\{\!\{\cdot\}\!\}$ to denote multisets.

# 3 The power of 0-dim persistent homology under color-based filtrations

In this section, we analyze the representational power of persistent homology when adopting color-based filtrations. We focus on the persistence of connected components (0-dimensional holes). We separately discuss vertex-color (Section 3.1) and edge-color (Section 3.2) filtrations, and then compare these approaches in Section 3.3. Proofs for all Lemmas and Theorems are in Appendix B.

## 3.1 Vertex-color filtrations

To help characterize the expressivity of persistent homology, we propose classifying persistence pairs $(f(x^{(b)}), f(x^{(d)}))$ as either *real holes, almost holes, or trivial holes*. In particular, if $f(x^{(d)}) \neq \infty$

and $f(x^{(b)}) \neq f(x^{(d)})$, we say the pair $(f(x^{(b)}), f(x^{(d)}))$ is an *almost hole*. If $f(x^{(b)}) = f(x^{(d)})$, the pair is called a *trivial hole*. Finally, we call $(f(x^{(b)}), f(x^{(d)}))$ a *real hole* if $f(x^{(d)}) = \infty$.

### 3.1.1 Real holes as connected components

Real holes denote topological features that persist until a filtration reaches the entire graph. Thus, real holes from 0-dim persistence diagrams are associated with connected components, regardless of the filtration. Regarding the relationship between filtrations and real holes, Lemma 2 establishes the necessary and sufficient condition for the existence of a filtration that produces persistence diagrams with distinct real holes. Such a condition is associated with the notion of component-wise colors. Formally, let $G$ and $G'$ be two graphs with connected components $C_1, \ldots, C_k$ and $C'_1, \ldots, C'_{k'}$, respectively. Also, let $X_i = \{c(v) \mid v \in V_{C_i}\}$ be the set of colors in $C_i$ and, similarly, $X'_i$ be the colors in $C'_i$. We say that $G$ and $G'$ have *distinct component-wise colors* if $\{\!\{X_i\}\!\}_{i=1}^{k} \neq \{\!\{X'_i\}\!\}_{i=1}^{k'}$.

**Lemma 2** (**Equivalence between component-wise colors and real holes**). *Let $G$ and $G'$ be two graphs. There exists some vertex-color filtration such that their persistence diagrams $\mathcal{D}_G^0$ and $\mathcal{D}_{G'}^0$ have different multisets of real holes iff $G$ and $G'$ have distinct component-wise colors.*

### 3.1.2 Almost holes as separating sets

Now we turn our attention to the characterization of almost holes. Our next result (Lemma 3) reveals the connection between almost holes and separating sets. Here, a *separating set $S$* of a graph $G$ is a subset of its vertices whose removal disconnects *some* connected component of $G$.

**Lemma 3** (**Almost holes and separating sets**). *Regarding the relationship between almost holes and separating sets, the following holds:*

1. *Let $(f(x^{(b)}), f(x^{(d)}))$ be an almost hole from a vertex-color filtration. Then the set $S = \{w \in V \mid f(c(w)) \geq f(x^{(d)})\}$ is a separating set of $G$.*

2. *Let $S$ be a separating set of $G$ that splits a connected component $C \subseteq G$ into $k$ components $C_1, C_2, \ldots, C_k$. Then, there exists a filtration that produces $k - 1$ almost holes if the set of colors of vertices in $\cup_{i=1}^{k} V_{C_i}$ is disjoint from those of the remaining vertices, i.e., $\{c(v) \mid v \in V \setminus \cup_{i=1}^{k} V_{C_i}\} \cap \{c(v) \mid v \in \cup_{i=1}^{k} V_{C_i}\} = \emptyset$.*

The relationship between almost holes and separating sets elucidated in Lemma 3 raises the question if we can use separating sets (obtained from colors) to compare almost holes across different graphs. The answer is no: even if the diagrams of two graphs only differ in their multisets of almost holes, the graphs might not have separating sets that split them into different numbers of components. For instance, consider the graphs in Figure 2, where numbers denote filter values. The persistence diagrams $\mathcal{D}_G^0 = \{\!\{(1, \infty), (2, 3), (2, 3), (3, 3), (3, 4), (4, 4)\}\!\}$ and $\mathcal{D}_{G'}^0 = \{\!\{(1, \infty), (2, 3), (2, 4), (3, 3), (3, 3), (4, 4)\}\!\}$ only differ in almost holes. Still, we cannot pick colors whose removal of associated vertices would result in different numbers of components.

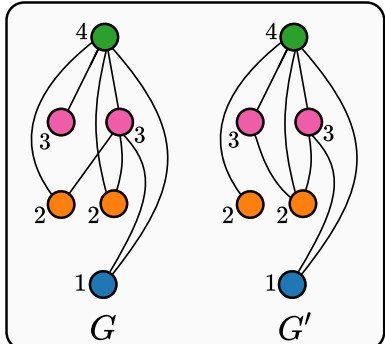

Figure 2: We cannot use color-based separating sets to compare almost holes across graphs. Although these filtrations produce different almost holes, there is no way to remove colors s.t. the resulting graphs have different numbers of components.

Next, we introduce the notion of color-separating sets (Definition 1). Importantly, Lemma 4 leverages this definition to characterize the graphs that can be distinguished based on almost holes. Specifically, it establishes that whenever the diagrams of two graphs differ in their multiset of almost holes, we can build a color-separating set.

**Definition 1** (**Color-separating sets**). *A color-separating set for a pair of graphs $(G, G')$ is a set of colors $Q$ such that the subgraphs induced by $V \setminus \{w \in V \mid c(w) \in Q\}$ and $V' \setminus \{w \in V' \mid c'(w) \in Q\}$ have distinct component-wise colors.*

We note that when $G$ and $G'$ have identical component-wise colors, the sets $\{w \in V \mid c(w) \in Q\}$ and $\{w \in V' \mid c'(w) \in Q\}$ induced by the color-separating set $Q$ are separating sets for $G$ and $G'$.

**Lemma 4** (**Distinct almost holes imply distinct color-separating sets**). *Let $\mathcal{D}_G^0$, $\mathcal{D}_{G'}^0$ be persistence diagrams for $G$ and $G'$. If the diagrams $\mathcal{D}_G^0$, $\mathcal{D}_{G'}^0$ differ in their multisets of almost holes, then there is a color-separating set for $G$ and $G'$.*

### 3.1.3  Bounds on the expressivity of vertex-color persistent homology

Regardless of the filtration, vertex-color PH always allows counting the numbers of connected components and vertices of a graph. If all vertices have the same color, then we cannot have any expressive power beyond $\beta^0$ and $|V|$ — when all vertices are added simultaneously, there cannot be almost holes as the finite living times of the holes are 0. Also, all real holes are identical, and we have $\mathcal{D}^0 = \{\!\{(1,\infty),\dots,(1,\infty),(1,1),...,(1,1)\}\!\}$, with $|\mathcal{D}^0| = |V|$.

For graphs with $m \geq 1$ colors, Lemma 5 shows that sets of birth times correspond to vertex colors. As a consequence, if the multisets of vertex colors differ for graphs $G$ and $G'$, then the corresponding persistence diagrams are also different in all filtrations.

**Lemma 5** (**Equivalence between birth times and vertex colors**). *There is a bijection between the multiset of birth times and the multiset of vertex colors in any vertex-color filtration.*

From Lemma 5, we can recover the multiset of colors from the persistence diagram and, consequently, distinguish graphs with different multisets. However, persistent homology uses vertex colors as input, and we do not need persistence diagrams to construct or compare such multisets. This highlights the importance of death times to achieve expressivity beyond identifying vertex colors. In fact, for non-trivial cases, the expressivity highly depends on the choice of filtration.

We have discussed the importance of color-separating sets (Lemma 4) and component-wise vertex colors (Lemma 2). With these notions, Theorem 1 formalizes the limits of expressivity that may be achieved with suitable filtration and characterize which pairs of graphs can, at best, be distinguished by comparing their persistence diagrams. Here, we only consider pairs of graphs that cannot be distinguished by their multisets of colors, as this corresponds to a trivial case.

**Theorem 1** (**The expressive power of vertex-color filtrations**). *For any two graphs $G$ and $G'$ with identical multisets of colors $\{\!\{c(v) : v \in V\}\!\} = \{\!\{c'(v) : v \in V'\}\!\}$, there exists a filtration such that $\mathcal{D}_G^0 \neq \mathcal{D}_{G'}^0$ if and only if there is a color-separating set for $G$ and $G'$.*

### 3.2  Edge-color filtrations

We now consider the expressivity of 0-dim persistent homology obtained from edge-color filtrations. The persistence diagrams are constructed exactly the same way. However, in this case, all holes are born at the same time (all vertices appear in $G_0$). This implies that all real holes are identical. Also, the diagrams do not contain trivial holes since $G_0$ does not have edges. All holes are either real holes or almost holes (of the form $(0,d)$). We also note that persistence diagrams will always have almost holes unless the graph is edgeless.

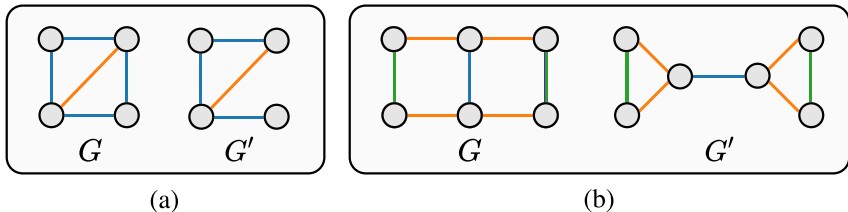

(a)  (b)

Figure 3: (a) $G$ and $G'$ differ in their multisets of colors, but no edge-color filtration can distinguish them. For instance, assume that $f(\text{'blue'}) = 1 < 2 = f(\text{'orange'})$. Then, $\mathcal{D}_G^0 = \mathcal{D}_{G'}^0 = \{\!\{(0,\infty),(0,1),(0,1),(0,1)\}\!\}$. The same holds for $f(\text{'blue'}) > f(\text{'orange'})$. (b) Graphs that have different disconnecting sets, but for which we can find filtrations that lead to identical diagrams.

Analogously to separating sets in vertex-color filtrations, Lemma 6 characterizes edge-based almost holes as *disconnecting sets* — a set of edges whose removal would increase the number of components.

**Lemma 6** (**Edge-based almost holes as disconnecting sets**). *Let $(0, f(x^{(d)}))$ be an almost hole from an edge-color filtration. Then $S = \{e \in E \mid f(l(e)) \geq f(x^{(d)})\}$ is a disconnecting set of $G$.*

Lemma 6 tells us how to construct a disconnecting set from an almost hole. Now, suppose we are given a disconnecting set $S$. Can we build an edge-color filtration for which $S$ can be obtained from

an almost hole? In other words, can we obtain a diagram with an almost hole $(0, f(x^{(d)}))$ such that $\{e \in E \mid f(l(e)) \geq f(x^{(d)})\}$ is equal to $S$? Lemma 7 shows that if the colors of edges in $S$ are distinct from those in $E \setminus S$, then there is a filtration that induces a persistence diagram with an almost hole from which we can reconstruct $S$.

**Lemma 7 (Reconstructing a disconnecting set).** *Let $G = (V, E, l, X)$ be a graph and $S \subseteq E$ be a disconnecting set for $G$. If the set of colors of $S$ is disjoint from that of $E \setminus S$, then there exists a filtration such that $S = \{e \in E \mid f(l(e)) \geq f(x^{(d)})\}$ for an almost hole $(0, f(x^{(d)})) \in \mathcal{D}^0$.*

### 3.2.1 Bounds on the expressivity of edge-color persistent homology

Similar to the vertex-color case, in any edge-color filtration, we have that $|\mathcal{D}^0| = |V|$ and the number of real holes is $\beta^0$. Also, the lowest expressivity is achieved when all edges have the same color. In this case, two graphs with different numbers of vertices or connected components have different persistence diagrams (and can be distinguished); otherwise, they cannot be distinguished.

We have seen in Lemma 5 that vertex-color filtrations encode colors as birth times. In contrast, birth times from edge-color filtrations are always trivially equal to zero. Thus, we cannot generalize Lemma 5 to edge-color filtrations. Instead, we can show there are graphs with different multisets of edge colors such that the graphs have the same persistence diagrams for any filtration (see Figure 3(a)).

Let us now consider lower limits for graphs with $m > 1$ edge colors. We can show that even if two graphs have different disconnecting sets (obtained from colors), there are filtrations that induce the same persistence diagrams. To see this behavior, consider the two graphs in Figure 3(b), where the deletion of blue edges disconnects one of the graphs but not the other. Although we can build an edge-color filtration that separates these graphs (i.e., $\mathcal{D}^0_G \neq \mathcal{D}^0_{G'}$), if we choose $f(\text{'green'}) = 3, f(\text{'orange'}) = 2$, and $f(\text{'blue'}) = 1$, we obtain $\mathcal{D}^0_G = D^0_{G'} = \{\!\{(0, \infty), (0, 1), (0, 2), (0, 2), (0, 2), (0, 2)\}\!\}$. Interestingly, even if two graphs have different sets of edge colors, we might still find filtrations that induce identical diagrams. The reason is that unlike vertex-color filtrations where trivial holes make sure that all vertices are represented in the diagrams, in edge-color filtrations there are no trivial holes. As a result, persistence diagrams from edge-color filtrations do not account for edges that do not lead to the disappearance of connected components.

Lemma 6 and Lemma 7 showed that edge-based almost holes can be characterized as disconnecting sets, somewhat analogously to vertex-based almost holes as separating sets. We complete the analogy by introducing the notion of color-disconnecting sets in Definition 2. We then use this notion to fully characterize the the expressive power of edge-color persistent homology in Theorem 2. More specifically, the existence of a color-disconnecting set between a given pair of graphs is a necessary and sufficient condition for distinguishing them based on 0-dimensional persistence diagrams.

**Definition 2 (Color-disconnecting sets).** *A color-disconnecting set for a pair of graphs $(G, G')$ is a set of colors $Q$ such that if we remove the edges of colors in $Q$ from $G$ and $G'$, we obtain subgraphs with different numbers of connected components.*

**Theorem 2 (The expressive power of edge-color filtrations).** *Consider two graphs $G$ and $G'$. There exists an edge-color filtration such that $\mathcal{D}^0_G \neq \mathcal{D}^0_{G'}$ if and only if there is a color-disconnecting set for $G$ and $G'$.*

### 3.3 Vertex-color versus edge-color filtrations

To compare vertex- and edge-color persistence diagrams, we consider graphs with vertex-coloring functions $c(\cdot)$ from which we derive edge-coloring ones $l(\cdot)$. In particular, for a graph $G = (V, E, c, X)$, its edge-coloring function $l : E \to X^2$ is defined as $l(v, w) = \{c(v), c(w)\}$.

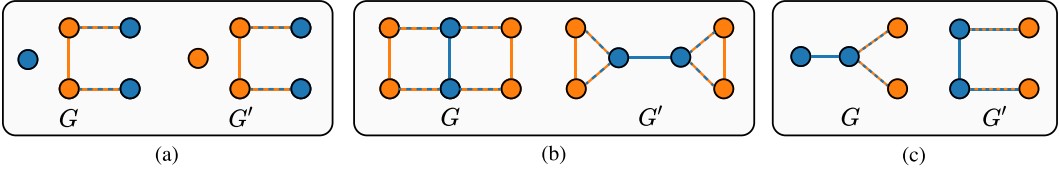

Figure 4: Illustration of graphs that cannot be distinguished based on (a) edge-color filtrations, (b) vertex-color filtrations, and (c) both vertex- and edge-color filtrations.

Recall that only vertex-color filtrations can 1) encode multisets of colors and 2) have real holes with different birth times. Naturally, we can find pairs of graphs $(G, G')$ for which we can obtain $\mathcal{D}_G^0 \neq \mathcal{D}_{G'}^0$ from vertex-color filtrations, but not from edge-color ones. Consider the graphs in Figure 4(a). The vertex-color filtration $f(\text{'blue'}) = 1, f(\text{'orange'}) = 2$ produces $\mathcal{D} = \{\!\{(1,\infty), (1,\infty), (1,2), (2,2), (2,2)\}\!\}$ and $\mathcal{D}' = \{\!\{(1,\infty), (1,1), (2,\infty), (2,2), (2,2)\}\!\}$. However, there is no edge-color filtration that would tell them apart — there are only two possible edge-color filtrations, leading to either $\mathcal{D} = \{\!\{(0,\infty), (0,\infty), (0,1), (0,2), (0,2)\}\!\} = \mathcal{D}'$, or $\mathcal{D} = \{\!\{(0,\infty), (0,\infty), (0,1), (0,1), (0,2)\}\!\} = \mathcal{D}'$.

We can also show that there are graphs that can be distinguished by edge-color filtrations but not by vertex-color ones. Intuitively, one can think of this as a result of edge colors being more fine-tuned. For instance, consider the graphs in Figure 4(b). We can separate these graphs using the function $f(\text{'orange'}) = 1, f(\text{'blue-orange'}) = 2$, and $f(\text{'blue'}) = 3$, which yields $\mathcal{D}_G^0 = \{\!\{(0,\infty), (0,1), (0,1), (0,2), (0,2), (0,3)\}\!\} \neq \{\!\{(0,\infty), (0,1), (0,1), (0,2), (0,2), (0,2)\}\!\} = \mathcal{D}_{G'}^0$. However, since there is no color-separating set for $G$ and $G'$, by Theorem 1, we have that $\mathcal{D}_G = \mathcal{D}_{G'}$ for all vertex-color filtrations. Theorem 3 formalizes the idea that none of the classes of color-based filtrations subsumes the other. In addition, Figure 4(c) illustrates that there are very simple non-isomorphic graphs that PH under both vertex- and edge-color filtrations cannot distinguish.

**Theorem 3** (**Edge-color vs. vertex-color filtrations**). *There exist non-isomorphic graphs that vertex-color filtrations can distinguish but edge-color filtrations cannot, and vice-versa.*

## 4  Going beyond persistent homology

We now leverage the theoretical results in Section 3 to further boost the representational capability of persistent homology. In particular, we propose modifying edge-color persistence diagrams to account for structural information that is not captured via the original diagrams. We call the resulting approach RePHINE (Refining PH by incorporating node-color into edge-based filtration). Notably, RePHINE diagrams are not only provably more expressive than standard color-based ones but also make 1-dimensional topological features redundant. Additionally, we show how to integrate RePHINE into arbitrary GNN layers for graph-level prediction tasks.

**Edge-color diagrams with missing holes.** A major drawback of edge-color filtrations is that information about the multisets of (edge) colors is lost, i.e., it cannot be recovered from persistence diagrams. To reconstruct disconnecting sets, we need the edge-color permutation given by the filtration function and the number of edges — both of which cannot be deduced from diagrams alone.

To fill this gap, we introduce the notion of *missing holes*. Conceptually, missing holes correspond to edges that are not associated with the disappearance of any connected component in a given filtration. By design, we set the birth time of missing holes to 1 — this distinguishes them from real and almost holes, which have birth times equal to 0. The death time of a missing hole corresponds to the first filtration step $f(x)$ that its corresponding edge color $x$ appears. We note that missing holes correspond to cycles obtained from 1-dim persistence diagrams.

As an example, consider the edge-color filtration in Figure 5, which produces $\mathcal{D}^0 = \{\!\{(0,\infty), (0,1), (0,2), (0,2), (0,4)\}\!\}$. We note that the orange edge and one of the orange-green ones do not 'kill' any 0-dim hole. This results in the missing holes $(1,3)$ and $(1,4)$. Clearly, missing holes bring in additional expressivity as, e.g., it would be possible to distinguish graphs that only differ in the orange edge in Figure 5. Still, edge-color diagrams with missing holes are not more expressive than vertex-color ones. For instance, they cannot separate the two graphs in Figure 3(a).

**Augmenting edge-color diagrams with vertex-color information.** To improve the expressivity of persistent homology, a simple approach is to merge tuples obtained from independent vertex- and edge-color filtrations. However, this would double the computational cost while only allowing distinguishing pairs of graphs that could already be separated by one of the filtrations. Ideally, we would like to go beyond the union of vertex- and edge-color persistence diagrams.

As in Section 3.3, we consider graphs with edge colors obtained from vertex-coloring functions. Also, we assume that $f_v$ and $f_e$ are independent vertex- and edge-color filtration functions, respectively. We propose adding two new elements to the tuples of edge-color diagrams with missing holes. Our augmented tuple is $(b, d, \alpha, \gamma)$ where $\alpha$ and $\gamma$ are the additional terms. Recall that, in any edge-color filtration, $G_0$ has $|V|$ connected components. Then, we can associate real or almost holes of edge-color diagrams with vertices in $G$. With this in mind, we define RePHINE diagrams as follows.

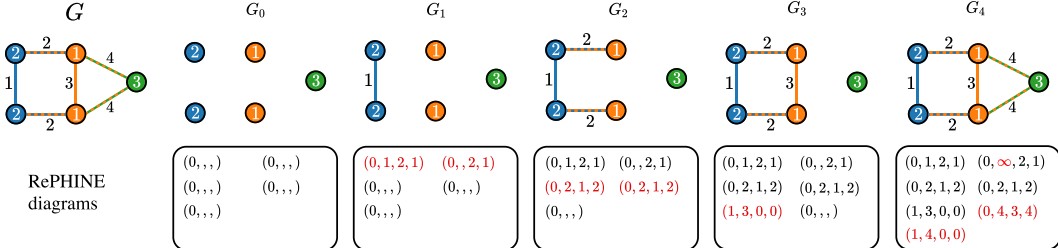

Figure 5: RePHINE diagrams. At $G_1$, one component dies and creates the almost hole $(0, 1, 2, 1)$. We also save that two nodes were discovered at $1$ (fourth component), with colors equal to $2$ (third component). At step 2, two other holes are killed, resulting in two tuples $(0, 2, 1, 2)$. At $G_3$, we obtain the missing hole $(1, 3, 0, 0)$. Finally, $G_4$ creates one almost hole and one missing hole.

**Definition 3** (**RePHINE diagram**). *The RePHINE diagram of a filtration on a graph $G$ is a multiset of cardinality $|V| + \beta_G^1$, with elements of form $(b, d, \alpha, \gamma)$. There are two cases:*

- *Case $b = 0$ (real or almost holes). Now, $b$ and $d$ correspond to birth and death times of a component as in edge-color filtration. We set $\alpha(w) = f_v(c(w))$ and $\gamma(w) = \min_{v \in \mathcal{N}(w)} f_e(\{\!\{c(w), c(v)\}\!\})$, where $w$ is the vertex that is associated with the almost or real hole. Vertices are matched with the diagram elements as follows: An almost hole (b,d) corresponds to an edge merging two connected components, $T_1, T_2$. Each of these connected components has exactly one vertex, $w_{T_1}$ or $w_{T_2}$, which has not yet been associated with any element of the RePHINE diagram. Let $w = \arg\max_{w' \in \{w_{T_1}, w_{T_2}\}} f_v(c(w'))$ , or if $f_v(c(w_{T_1})) = f_v(c(w_{T_2}))$, then $w = \arg\min_{w' \in \{w_{T_1}, w_{T_2}\}} \gamma(w')$. The vertices that are associated with real holes are vertices that have not died after the last filtration step.*
- *Case $b = 1$ (missing holes). Here, the entry $d$ is the filtration value of an edge $e$ that did not kill a hole but produces a cycle that appears at the filtration step associated with adding the edge $e$. The entries $\alpha$ and $\gamma$ take uninformative values (e.g., 0).*

Figure 5 provides an example of RePHINE diagrams. Further details of the procedure can be found in Appendix C. Notably, our scheme can be computed efficiently at the same cost as standard persistence diagrams and is consistent — we obtain identical diagrams for any two isomorphic colored graphs.

**Theorem 4** (**RePHINE is isomorphism invariant**). *Let $G$, $G'$ be isomorphic graphs. Then, any edge-color and vertex-color filtrations produce identical RePHINE diagrams for $G$ and $G'$.*

In addition, Theorem 5 shows that RePHINE diagrams are strictly more expressive than those from both vertex- and edge-color filtrations, including 0- and 1-dim topological features. Figure 4(c) provides an example of graphs that cannot be recognized by any color-based filtration, but for which we can obtain distinct RePHINE diagrams.

**Theorem 5** (**RePHINE is strictly more expressive than color-based PH**). *Let $\mathcal{D}, \mathcal{D}'$ be the persistence diagrams associated with any edge or vertex-color filtration of two graphs. If $\mathcal{D} \neq \mathcal{D}'$, then there is a filtration that produces different RePHINE diagrams. The converse does not hold.*

Despite its power, there are simple non-isomorphic graphs RePHINE cannot distinguish. In particular, if two graphs have one color, RePHINE cannot separate graphs of equal size with the same number of components and cycles. For example, star and path graphs with 4 vertices of color $c_1$ produce identical RePHINE diagrams of the form $\{\!\{(0, d, a, d), (0, d, a, d), (0, d, a, d), (0, \infty, a, d)\}\!\}$, where $d = f_e(\{\!\{c_1, c_1\}\!\})$ and $a = f_v(c_1)$ for arbitrary edge- and vertex-color filtration functions.

**Combining RePHINE and GNNs.** RePHINE diagrams can be easily incorporated into general GNN layers. For instance, one can follow the scheme in [15] to combine non-missing hole information with node features and leverage missing holes as graph-level attributes. However, here we adopt a simple scheme that processes RePHINE tuples using DeepSets [38]. These topological embeddings are then grouped using a pooling layer and concatenated with the graph-level GNN embedding. The resulting representation is fed to a feedforward network to obtain class predictions. Formally, let $\mathcal{N}_G(u)$ denote the set of neighbors of vertex $u$ in $G$, and $h_u^{(0)} = c(u)$ for all $u \in V$. We compute

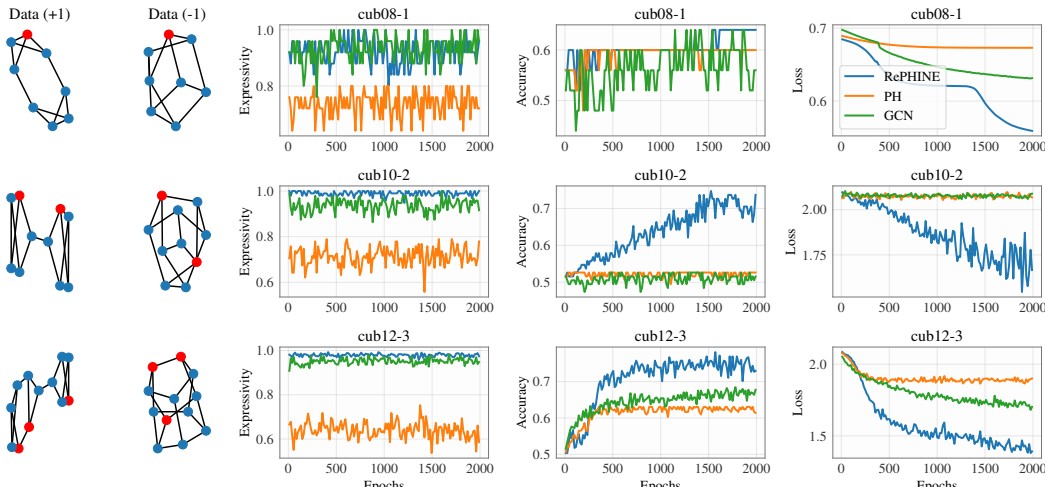

Figure 6: Average learning curves for RePHINE, PH, and GCN on connected cubic graphs. RePHINE can learn representations in cases where PH and GNNs struggle to capture structural information. RePHINE shows better expressivity and fitting capability on `Cub10-2` and `Cub12-3`.

GNN and RePHINE embeddings (denoted by $r^{(\ell)}$) at layer $\ell$ recursively as:

$$\tilde{h}_u^{(\ell)} = \text{AGG}^{(\ell)}(\{\!\{h_w^{(\ell-1)} \mid w \in \mathcal{N}_G(u)\}\!\}) \quad \forall u \in V \qquad \mathcal{R}^{(\ell)} = \text{RePHINE}(f_v^{(\ell)}, f_e^{(\ell)}, \{\!\{h_u^{(\ell)}\}\!\}_{u \in V})$$

$$h_u^{(\ell)} = \text{UPDATE}^{(\ell)}\left(h_u^{(\ell-1)}, \tilde{h}_u^{(\ell)}\right) \quad \forall u \in V \qquad r^{(\ell)} = \phi^{(\ell)}(\sum_{d \in \mathcal{R}^{(\ell)}} \psi^{(\ell)}(d))$$

where $f_v^{(\ell)}, f_e^{(\ell)}, \psi^{(\ell)}, \phi^{(\ell)}, \text{AGG}^{(\ell)}$, and $\text{UPDATE}^{(\ell)}$ are arbitrary non-linear mappings, usually implemented as feedforward neural nets. After $L$ layers, we obtain the combined RePHINE-GNN graph-level representation as $[\text{POOL}_1(\{r^{(\ell)}\}_\ell) \parallel \text{POOL}_2(\{h_u^{(L)}\}_u)]$, where $\text{POOL}_1$ is either mean or concatenation, and $\text{POOL}_2$ is an order invariant operation.

## 5  Experiments

In this section, we compare RePHINE to standard persistence diagrams from an empirical perspective. Our main goal is to evaluate whether our method enables powerful graph-level representation, confirming our theoretical analysis. Therefore, we conduct two main experiments. The first one leverages an artificially created dataset, expected to impose challenges to persistent homology and MP-GNNs. The second experiment aims to assess the predictive performance of RePHINE in combination with GNNs on popular benchmarks for graph classification. All methods were implemented in PyTorch [31], and our code is available at https://github.com/Aalto-QuML/RePHINE.

**Synthetic data.** We consider three datasets of cubic graphs (or 3-regular graphs): cub08, cub10, and cub12 [6]. These graphs cannot be distinguished by 1-WL and color-based PH as all vertices share the same color. Thus, we modify the datasets by changing the colors of 1, 2, or 3 vertices in each graph sample, resulting in the modified datasets `cub08-1`, `cub10-2`, and `cub12-3`. Also, we randomly partition each dataset and create a balanced binary classification task. We expect this to keep the hardness of the task while allowing some distinguishability.

We compare standard 0-dim persistence diagrams from vertex-color filtrations (referred to as PH) to 0-dim RePHINE (i.e., no missing holes). Both approaches are processed using `DeepSets` with exactly the same structure and optimization procedure. Also, they operate on the original colors, not on GNN embeddings. For completeness, we report results for a 2-layer graph convolutional network (GCN) [22] followed by an MLP. We are interested in assessing if the persistence modules can overfit the observed graphs. We also monitor if the methods obtain different representations for each graph, measured in terms of the proportion of unique graph embeddings over training (which we call *expressivity*). We provide further details and additional results with 1-dim persistence diagrams in the supplementary material.

Table 1: Predictive performance on graph classification. We denote in bold the best results. For ZINC, lower is better. For most datasets, RePHINE is the best-performing method.

| GNN | Diagram | NCI109 ↑ | PROTEINS ↑ | IMDB-B ↑ | NCI1 ↑ | MOLHIV ↑ | ZINC ↓ |
|-----|---------|----------|------------|----------|--------|----------|--------|
| GCN | - | $76.46 \pm 1.03$ | $70.18 \pm 1.35$ | $64.20 \pm 1.30$ | $74.45 \pm 1.05$ | $74.99 \pm 1.09$ | $0.875 \pm 0.009$ |
| | PH | $77.92 \pm 1.89$ | $69.46 \pm 1.83$ | $64.80 \pm 1.30$ | $79.08 \pm 1.06$ | $73.64 \pm 1.29$ | $0.513 \pm 0.014$ |
| | RePHINE | $\mathbf{79.18} \pm 1.97$ | $\mathbf{71.25} \pm 1.60$ | $\mathbf{69.40} \pm 3.78$ | $\mathbf{80.44} \pm 0.94$ | $\mathbf{75.98} \pm 1.80$ | $\mathbf{0.468} \pm 0.011$ |
| GIN | - | $76.90 \pm 0.80$ | $\mathbf{72.50} \pm 2.31$ | $\mathbf{74.20} \pm 1.30$ | $76.89 \pm 1.75$ | $70.76 \pm 2.46$ | $0.621 \pm 0.015$ |
| | PH | $78.35 \pm 0.68$ | $69.46 \pm 2.48$ | $69.80 \pm 0.84$ | $79.12 \pm 1.23$ | $73.37 \pm 4.36$ | $0.440 \pm 0.019$ |
| | RePHINE | $\mathbf{79.23} \pm 1.67$ | $72.32 \pm 1.89$ | $72.80 \pm 2.95$ | $\mathbf{80.92} \pm 1.92$ | $\mathbf{73.71} \pm 0.91$ | $\mathbf{0.411} \pm 0.015$ |

Figure 6 shows the learning curves for 2000 epochs, averaged over five runs. Notably, for all datasets, the expressivity of RePHINE is significantly higher than those from PH and similar to GNN's. On cub10-2, while PH and GNN obtain accuracies of around 0.5, RePHINE allows a better fit to the observed data, illustrated by higher accuracy and lower loss values.

**Real-world data.** To assess the performance of RePHINE on real data, we use six popular datasets for graph classification (details in the Supplementary): PROTEINS, IMDB-BINARY, NCI1, NCI109, MOLHIV and ZINC [7, 16, 20]. We compare RePHINE against standard vertex-color persistence diagrams (simply called PH here). Again, we do not aim to benchmark the performance of topological GNNs, but isolate the effect of the persistence modules. Thus, we adopt *default* (shallow) GNN architectures and process the persistence diagrams exactly the same way using DeepSets. We report the mean and standard deviation of predictive metrics (AUC for MOLHIV, MAE for ZINC, and Accuracy for the remaining) over five runs. We provide further implementation details in Appendix C.

Table 1 shows the results of PH and RePHINE combined with GCN [22] and GIN [37] models. Notably, RePHINE consistently outperforms PH, being the best-performing method in 10 out of 12 experiments. Overall, we note that GIN-based approaches achieve slightly better results. Our results suggest that RePHINE should be the default choice for persistence descriptors on graphs.

Table 2: PersLay vs. RePHINE: Accuracy results on graph classification.

| Method | NCI109 | PROTEINS | IMDB-B | NCI1 |
|--------|--------|----------|--------|------|
| PersLay | $70.12 \pm 0.83$ | $67.68 \pm 1.94$ | $68.60 \pm 5.13$ | $68.86 \pm 0.86$ |
| RePHINE+Linear | $\mathbf{73.27} \pm 1.69$ | $\mathbf{71.96} \pm 1.85$ | $\mathbf{70.40} \pm 2.97$ | $\mathbf{74.94} \pm 1.35$ |

**Comparison to PersLay [2].** We also compare our method against another topological neural network, namely, PersLay. Since PersLay does not leverage GNNs, we adapted our initial design for a fair comparison. Specifically, we compute RePHINE diagrams with learned filtration functions and apply a linear classifier to provide class predictions. Also, we concatenate the vectorial representations of the RePHINE diagrams with the same graph-level features obtained using PersLay. We refer to our variant as RePHINE+Linear. Table 2 reports accuracy results over 5 runs on 4 datasets. For all datasets, RePHINE+Linear achieves higher accuracy, with a significant margin overall.

## 6 Conclusion, Broader Impact, and Limitations

We resolve the expressivity of persistent homology methods for graph representation learning, establishing a complete characterization of attributed graphs that can be distinguished with general node- and edge-color filtrations. Central to our analyses is a novel notion of color-separating sets.

Much like how WL test has fostered more expressive graph neural networks (GNNs), our framework of color-separating sets enables the design of provably more powerful topological descriptors such as RePHINE (introduced here). RePHINE is computationally efficient and can be readily integrated into GNNs, yielding empirical gains on several real benchmarks.

We have not analyzed here other types of filtrations, e.g., those based on the spectral decomposition of graph Laplacians. Future work should also analyze the stability, generalization capabilities, and local versions of RePHINE. Overall, we expect this work to spur principled methods that can leverage both topological and geometric information, e.g., to obtain richer representations for molecules in applications such as drug discovery and material design.

## Acknowledgments

This work was supported by Academy of Finland (Flagship programme: Finnish Center for Artificial Intelligence FCAI) and a tenure-track starting grant by Aalto University. We also acknowledge the computational resources provided by the Aalto Science-IT Project from Computer Science IT. We are grateful to the anonymous area chair and reviewers for their constructive feedback. Johanna Immonen thanks Tuan Anh Pham, Jannis Halbey, Negar Soltan Mohammadi, Yunseon (Sunnie) Lee, Bruce Nguyen and Nahal Mirzaie for their support and for all the fun during her research internship at Aalto University in the summer of 2022.

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

# Supplementary material: Going beyond persistent homology using persistent homology

## A  Persistent homology

Persistent homology (PH) is one of the workhorses for topological data analysis (TDA). A central idea underlying PH is to investigate the multiresolution structure in data through the lens of low-dimensional topological features such as connected components (0-dimensional), loops (1-dimensional), and voids (2-dimensional). Here, we provide a brief description of PH, and how it extends to graphs. In particular, we do not present proofs and do not show that the constructions are well-defined. For a detailed treatment, we refer the reader to [13], [9].

We will first define homological groups. They allow to characterise p-dimensional holes in a topological space such as a simplicial complex. We present the theory for simplicial complexes, as our focus is on 1-dimensional simplicial complexes (i.e. graphs).

Let $K$ be a simplicial complex. The $p$-chains are formal sums $c = \sum a_i \sigma_i$, where $a_i \in \mathbb{Z}/2\mathbb{Z}$ and $\sigma_i$ are $p$-simplices in $K$. One can think of $p$-chain as a set of $p$-simplices such that $a_i = 1$. Together with componentwise addition, $p$-chains form the group $C_p(K)$.

Now, consider a simplex $\sigma = (v_0, ..., v_p) \in K$. We can define a boundary for $\sigma$ by

$$\partial_p \sigma = \sum_{j=0}^{p} (v_0, ..., v_{j-1}, v_{j+1}, ..., v_p),$$

i.e., $\partial_p \sigma$ is a sum of the $(p-1)$-dimensional faces of $\sigma$. We can extend this to define a boundary homomorphism $\partial_p : C_p(K) \to C_{p-1}(K)$ where $\partial_p \sum a_i \sigma_i = \sum a_i \partial_p \sigma_i$. Thus, we can define a sequence of groups

$$...C_{p+1}(K) \xrightarrow{\partial_{p+1}} C_p(K) \xrightarrow{\partial_p} C_{p-1}(K)...,$$

each connected with a boundary homomorphism. This sequence is chain complex, and it is the last definition we need in order to consider homology groups.

The $p$th homology group is a group of $p$-chains with empty boundary (*i.e.* $\partial_p \sigma = 0$) such that each of these particular $p$-chains (cycles) are a boundary of a different simplex in $C_{p+1}(K)$. So, we can define the homology group as the quotient space

$$H_p = \ker \partial_p / \mathrm{Im} \partial_{(p+1)}.$$

The rank of $H_p$ is equal to the $p$th *Betti* number ($\beta_p$). Then, let us see how the homology groups can be refined to gain persistent homology groups.

Persistent homology tracks the evolution of *Betti* numbers in a sequence of chain complexes. For this, we need a filtration, which is an increasing sequence of simplicial complexes $(\mathcal{F}_i)_{i=1}^r$ such that $\mathcal{F}_1 = \emptyset \subseteq \mathcal{F}_2 \subseteq \ldots \subseteq \mathcal{F}_r = K$. By constructing all homology groups for each of these simplicial complexes, we can capture changes. New holes (or, homology classes) may emerge, or they may be annihilated such that only the older remains. As such, we can associate a pair of timestamps, or persistence points, $(i, j)$ for every hole to indicate the filtration steps it appeared and disappeared. The persistence of a point $(i, j)$ is the duration for which the corresponding feature was in existence, i.e., the difference $|i - j|$. We set $j = \infty$ if the hole does not disappear, i.e. is present at the last filtration step. The extension to persistent homology groups and persistent *Betti* numbers is natural:

$$H_p^{i,j} = \ker \partial_p / (\mathrm{Im} \partial_{(p+1)} \cap \ker \partial_p),$$

and the $p$th persistent *Betti* number $\beta_p^{i,j}$ are given by the rank of $H_p^{i,j}$ as earlier. Lastly, a persistent diagram that consists of the persistent points $(i, j)$ with multiplicities

$$\mu_p^{i,j} = (\beta_p^{i,j-1} - \beta_p^{i,j}) - (\beta_p^{i-1,j-1} - \beta_p^{i-1,j})$$

where $i < j$, encodes the persistent homology groups entirely by the Fundamental Lemma of Persistent Homology.

For graphs, the filtration may be viewed as creating an increasing sequence of subgraphs. This entails selecting a subset of vertices and edges of the graph at each step of the filtration. One can learn a parameterized function $f$ (e.g., a neural network) to assign some value to each $\sigma \in K$, and thereby select the subsets $K_i$ based on a threshold $\alpha_i \in \mathbb{R}$. That is, $f$ induces a filtration $(\mathcal{F}_i)_{i=1}^r$ using a sequence $(\alpha_i)_{i=1}^r$ such that $\alpha_1 \geq \alpha_2 \geq \ldots \geq \alpha_r$:

$$\mathcal{F}_i \triangleq \mathcal{F}(f; \alpha_i) = \{\sigma \in K : f(\sigma) \geq \alpha_i\}.$$

We provide a detailed pseudocode in Algorithm 1 to compute the persistence diagram for an input graph. The algorithm uses the Union-Find data structure, also known as a disjoint-set forest. The code assumes we are given vertex-color filter values, stored in the variable vValues. The algorithm returns a multiset containing 0- and 1-dimensional persistence tuples (i.e., persistence diagrams).

---

**Algorithm 1** Computing persistence diagrams

---

**Require:** $V, E,$ vValues             ▷ Vertices, edges, and vertex-color filter values
  uf ← UNIONFIND($|V|$)
  pers0 ← zeros($|V|, 2$)                             ▷ Initialize the persistence tuples
  pers1 ← zeros($|E|, 2$)
  **for** $e \in$ E **do**
     $(v, w) \leftarrow e$
     eValues$[e]$ ← max(vValues$[v]$, vValues$[w]$)
  **end for**
  pers0$[:, 1]$ ← vValues                          ▷ Pre-set the 'birth' times
  SINDICES, SVALUES ← SORT(eValues)
  **for** $e,$ weight $\in$ Pair(SINDICES, SVALUES) **do** ▷ Pair is equivalent to the zip function in Python
     $(v, w) \leftarrow e$
     younger ← uf.find($v$)                   ▷ younger denotes the component that will die
     older ← uf.find($w$)
     **if** younger = older **then**                        ▷ A cycle was detected
       pers1$[e, 1]$ ← weight
       pers1$[e, 2]$ ← $\infty$
       **continue**
     **else**
       **if** vValues[younger] < vValues[older] **then**
         younger, older, $v, w$ ← older, younger, $w, v$
       **end if**
     **end if**
     pers0[younger, 2] ← weight
     uf.merge($v, w$)                         ▷ Merge two connected components
  **end for**
  **for** $r \in$ uf.roots() **do**
     pers0$[r, 2]$ ← $\infty$
  **end for**
  $\mathcal{D}_v$ ← JOIN(pers0, pers1)
  **return** $\mathcal{D}_v$

---

# B Proofs

## B.1 Proof of Lemma 1: Vertex-based filtrations can generate inconsistent diagrams

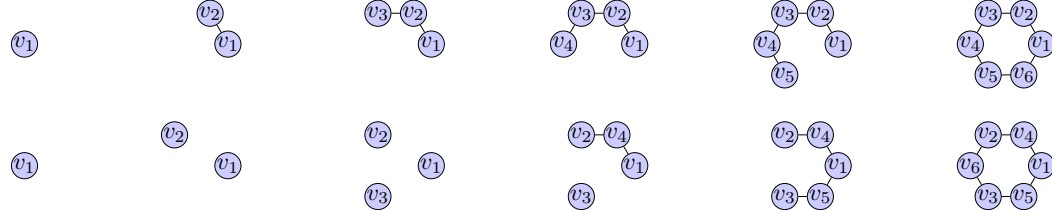

Figure S1: Filtrations induced by injective vertex filter functions for two isomorphic graphs. Node ids are used as filter function. The top-row filtration induces the persistence diagram $\mathcal{D}_1^0 = \{\!\{(1,\infty),(2,2),(3,3),(4,4),(5,5),(6,6)\}\!\}$, while the second-row filtration produces $\mathcal{D}_2^0 = \{\!\{(1,\infty),(2,4),(3,5),(4,4),(5,5),(6,6)\}\!\}$.

*Proof.* Consider a simple cyclic graph with 6 vertices that share the same color. Since the vertices are structurally identical and have the same color, one would expect to get a single persistence diagram irrespective of the labeling of the vertices. However, this is not the case. Consider two different labelings for the vertices on the graph: $\ell_1 = (v_1,v_2,v_3,v_4,v_5,v_6)$ and $\ell_2 = (v_1,v_4,v_2,v_6,v_3,v_5)$ (see Figure S1). Now, consider an injective vertex-based filtration where $f(v_i) > f(v_j)$ if $i > j$. Then, we obtain two different persistence diagrams, $\mathcal{D}_1 = \{\!\{(1,\infty),(2,2),(3,3),(4,4),(5,5),(6,6)\}\!\}$ and $\mathcal{D}_2 = \{\!\{(1,\infty),(2,4),(3,5),(4,4),(5,5),(6,6)\}\!\}$. We note that for any choice of vertex-based injective filter function on this cycle graph, we can follow a similar procedure to build two different labelings such that the persistence diagrams are different. $\qquad\square$

## B.2 Proof of Lemma 2: Equivalence between component-wise colors and real holes

*Proof.* We consider two arbitrary graphs $G = (V,E,c,X)$ and $G' = (V',E',c',X')$ and an injective filter function $f : X \cup X' \to \mathbb{R}$. We note that if $G$ and $G'$ do not have the same number of connected components (i.e., $\beta_G^0 \neq \beta_{G'}^0$), then $G$ and $G'$ differ on the number of real holes, i.e., their multisets of real holes are different trivially. Thus, we now assume $\beta_G^0 = \beta_{G'}^0 = k$. We also assume both graphs have same colors — if there is a color in $G$ that is not in $G'$, the claim is trivial.

[$\Rightarrow$] Recall that $X_i = \{c(v) \mid v \in V_{C_i}\}$ denotes the set of colors in the component $C_i \subseteq G$. Similarly, $X_i'$ is the set of colors in $C_i' \subseteq G'$. We want to show that if $\{\!\{X_i\}\!\}_{i=1}^k \neq \{\!\{X_i'\}\!\}_{i=1}^k$, then there exists a filtration such that the multisets of real holes are different. We proceed with a proof by induction on the number of colors.

If there is only 1 color, component-wise colors cannot differ for graphs with $\beta_G^0 = \beta_{G'}^0$. Let us thus consider 2 colors (say, $b$ and $w$). For 2 colors, there are only three possibilities for what $X_h \in \{\!\{X_i\}\!\}_{i=1}^k$ may be: $\{b\}, \{w\}$ or $\{b,w\}$. Now, let us denote the multiplicities of $\{b\}, \{w\}$ and $\{b,w\}$ in $\{\!\{X_i\}\!\}_{i=1}^k$ by $n_1, n_2$ and $n_3$, respectively. Note that for $G$ and $G'$ with $\beta_G^0 = \beta_{G'}^0$, we have $n_1 + n_2 + n_3 = n_1' + n_2' + n_3'$. Thus, when $\{\!\{X_i\}\!\}_{i=1}^k \neq \{\!\{X_i'\}\!\}_{i=1}^k$, there are four cases to consider:

1. $n_1 \neq n_1', n_2 \neq n_2', n_3 = n_3'$: Here, $n_2 + n_3 \neq n_2' + n_3'$ correspond to multiplicities of real holes $(w,\infty)$ for $G$ and $G'$ respectively, in a filtration that introduces the color $w$ first.

2. $n_1 \neq n_1', n_2 = n_2', n_3 \neq n_3'$ : Again, $n_2 + n_3 \neq n_2' + n_3'$ correspond to multiplicities of real holes $(w,\infty)$ for $G$ and $G'$ respectively in a filtration that introduces the color $w$ first.

3. $n_1 = n_1', n_2 \neq n_2', n_3 \neq n_3'$: Now, $n_1 + n_3 \neq n_1' + n_3'$ correspond to multiplicities of real holes $(b,\infty)$ for $G$ and $G'$ respectively in a filtration that introduces the color $b$ first.

4. $n_1 \neq n_1', n_2 \neq n_2', n_3 \neq n_3'$: Similarly, $n_1 + n_3 \neq n_1' + n_3'$ correspond to multiplicities of real holes $(b,\infty)$ for $G$ and $G'$ respectively in a filtration that introduces the color $b$ first.

Note that cases as $n_1 \neq n_1', n_2 = n_2', n_3 = n_3'$ are not possible as $n_1 + n_2 + n_3 = n_1' + n_2' + n_3'$.

Let us then assume that there are $l$ colors, and there exists a permutation of the colors $\{c_1, c_2, ..., c_l\}$ that induces a filtration giving different colored representatives.

Let us consider graphs $G$ and $G'$ with $l+1$ colors. Now, if $\{\!\{X_i\}\!\}_{i=1}^k \neq \{\!\{X_i'\}\!\}_{i=1}^k$ for subgraphs of $G$ and $G'$ with only $l$ colors, the permutation $\{c_{l+1}, c_1, c_2, ..., c_l\}$ induces a filtration where the representatives of first $l$ colors differ (and there may or may not be a difference also in the representatives of the $l+1$-th color). However, if there are no such subgraphs, this means that each of the pairs of unmatched component-colors contain the $l+1$ th color. Now $\{c_1, c_2, ..., c_l, c_{l+1}\}$ must induce the wanted kind filtration, since now the representatives of each component are as in $l$ colors. The claim follows by the induction principle.

$[\Leftarrow]$ Now, we want to prove that if there is a filtration such that the multisets of real holes differ, then $\{\!\{X_i\}\!\} \neq \{\!\{X_i'\}\!\}$. We proceed with a proof by contrapositive.

Assume that $\{\!\{X_i\}\!\} = \{\!\{X_i'\}\!\}$. Recall that, for a filter $f$, the color of the representatives of a real hole associated with $C_i$ is given by $\arg\min_{x \in X_i} f(x)$. If $\{\!\{X_i\}\!\} = \{\!\{X_i'\}\!\}$, it implies that the multisets of colors of the representatives are identical. Finally, note that the birth times of real holes are functions of these colors and, therefore, are identical as well. $\square$

### B.3 Proof of Lemma 3: Almost holes and separating sets

**Statement 1:** We want to show that if $(f(x^{(b)}), f(x^{(d)}))$ is an almost hole, then $S = \{v \in V | f(c(v)) \geq f(x^{(d)})\}$ is a separating set of $G = (V, E, c, X)$.

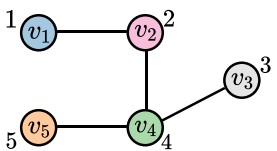

Figure S2: Graph to help illustrate the connectivity of almost holes.

*Proof.* Let $d = (f(x^{(b)}), f(x^{(d)}))$ be an almost hole. Then, we know there is at least one vertex $w$ of color $c(w) = x^{(b)}$ that gives birth to a new connected component at the filtration step $G_{f(x^{(b)})}$. Also, there is a distinct vertex $w'$ such that $w$ and $w'$ are not in the same component at $G_{f(x^{(b)})}$ but are connected at $G_{f(x^{(d)})}$. The existence of $w'$ is guaranteed since if there was no such $w'$ that gets connected to $w$ at $G_{f(x^{(d)})}$, $d$ would be a real hole, or if $w$ was connected to all other nodes at $G_{f(x^{(b)})}$, $d$ would be a trivial hole. Figure S2 illustrates a filtration on a 5-vertex graph with 5 colors. The filtration produces the persistence diagram $\{\!\{(1, \infty), (2, 2), (3, 4), (4, 4), (5, 5)\}\!\}$, with a single almost hole $(3, 4)$. According to our description, $w$ corresponds to $v_3$ (with $x^{(b)} = $ 'grey' and $f(x^{(b)}) = 3$), and $v_1$ could be a candidate to $w'$, for instance.

The discovery of the vertices in $T = \{v \in V \mid f(c(v)) = f(x^{(d)})\}$ connects $w$ to $w'$ since this set is added at the step when the component associated with $w$ dies at $f(x^{(d)})$. Equivalently, $T$ is a separating set of $G_{f(x^{(d)})}$. However, we want a separating set of $G$ (not of $G_{f(x^{(d)})}$). Finally, we note that expanding $T$ to $S = \{v \in V \mid f(v) \geq f(x^{(d)})\}$ suffices to obtain a separating set of $G$. $\square$

**Statement 2:** Let $S$ be a separating set of $G$ that splits a connected component $C \subseteq G$ into $k$ components $C_1, C_2, \ldots, C_k$. Then, there exists a filtration that produces $k-1$ almost holes if the set of colors of vertices in $\cup_{i=1}^k V_{C_i}$ is disjoint from those of the remaining vertices, i.e., $\{c(v) \mid v \in V \setminus \cup_{i=1}^k V_{C_i}\} \cap \{c(v) \mid v \in \cup_{i=1}^k V_{C_i}\} = \emptyset$.

*Proof.* Let us denote by $C_1, C_2, ..., C_k$ the connected components that $S$ separates $C$ into. We can first set a restriction $f|_{\cup_{i=1}^k V_{C_i}}$ to be any function mapping vertex colors to $\{1, 2, ..., |\cup_{i=1}^k V_{C_i}|\}$ — i.e., vertices in $\cup_{i=1}^k V_{C_i}$ must take filtration values in $\{1, 2, ..., |\cup_{i=1}^k V_{C_i}|\}$. Similarly, we can set $f|_{V \setminus \cup_{i=1}^k V_{C_i}}$ to be any function to $\{|\cup_{i=1}^k V_{C_i}| + 1, ..., |V|\}$.

The function $f$ obtained by combining the domains of $f|_{\cup_{i=1}^k V_{C_i}}$ and $f|_{V \setminus \cup_{i=1}^k V_{C_i}}$ is well defined due to the assumption $\{c(v) \mid v \in V \setminus \cup_{i=1}^k V_{C_i}\} \cap \{c(v) \mid v \in \cup_{i=1}^k V_{C_i}\} = \emptyset$. Since $C_1, C_2 \ldots$, $C_k$ are not path-connected, the persistence diagram induced by $f$ must have $k$ holes that are born at filtration steps in $\{1, 2, ..., |\cup_{i=1}^k V_{C_i}|\}$. Also, since the vertices of $S$ are added at filtration steps in $\{|\cup_{i=1}^k V_{C_i}| + 1, ..., |V|\}$, all holes die, forcing the birth and death times to be different. Thus, there must be one real hole corresponding to the connected component $C$ and $k-1$ almost holes. $\square$

### B.4 Proof of Lemma 4: Distinct almost holes imply distinct color-separating sets

*Proof.* We will consider two cases. The first one assumes that the multisets of real holes of $\mathcal{D}_G$ and $\mathcal{D}_{G'}$ are different. In the second case, we consider identical multisets of real holes and different multisets of almost holes.

**Case 1: multisets of real holes differ**. By Lemma 2, we have that the graphs have distinct component-wise colors - that is, an empty set is a color-separating set.

**Case 2: $\mathcal{D}_G^0$ and $\mathcal{D}_{G'}^0$ have identical real holes, but different multisets of almost holes**. We want to show that there is a color-separating set for $G$ and $G'$. We note that we can split the condition of distinct multisets of almost holes into two sub-cases: (i) There is some color $x_0$ such that there are more almost holes with birth time $f(x_0)$ in $\mathcal{D}_G^0$ than in $\mathcal{D}_{G'}^0$; (ii) There is some color $x_0$ such that there are more almost holes with death $f(x_0)$ in $\mathcal{D}_G^0$ than in $\mathcal{D}_{G'}^0$.

Let us first consider case (i). By the definition of birth time, we have that $G_{f(x_0)}$ has more connected components of color set $\{x_0\}$ than $G'_{f(x_0)}$. As such, $\{x \in X \cup X' | f(x) > f(x_0)\}$ is a color-separating set for $G$ and $G'$.

For case (ii), we assume that there are equally many births of almost holes associated to the the color $x_0$ — otherwise we return to case (i), for which we showed how to build a color-separating set. We note that if there is a different number of connected components at any earlier filtration step than when $x_0$ is introduced (i.e. $f(y) < f(x_0)$), then $\{x \in X \cup X' | f(x) > f(y)\}$ is a color separating set — since $G_{f(y)}$ and $G'_{f(y)}$ do not have as many connected components, they cannot have identical component-wise colors. However, if there is no such filtration step $f(y)$, it follows that $G_{f(x_0)}$ and $G'_{f(x_0)}$ cannot have the same number of components. This follows since vertices of color $x_0$ kill more connected components in $G_{f(x_0)}$ than in $G'_{f(x_0)}$, while prior to this, the numbers of components were equal. Therefore, $\{x \in X \cup X' | f(x) > f(x_0)\}$ is a color-separating set. $\square$

### B.5 Proof of Lemma 5: Equivalence between birth times and vertex colors

*Proof.* We consider a graph $G = (V, E, c, X)$ and any injective vertex-color filter $f : X \to \mathbb{R}$ from which we obtain a persistence diagram $\mathcal{D}^0$. We want to show that there exists a bijection between the multiset of birth times $\mathcal{B} = \{\!\{b \mid (b, d) \in \mathcal{D}^0\}\!\}$ and the multiset of vertex colors $\mathcal{X} = \{\!\{c(v) \mid v \in V\}\!\}$. Note that we can also represent a multiset as a pair $\mathcal{B} = (S_\mathcal{B}, m_\mathcal{B})$ where $S_\mathcal{B}$ is a set comprising the distinct elements of $\mathcal{B}$, and $m_\mathcal{B} : S_\mathcal{B} \to \mathbb{N}$ is a multiplicity function that gives the number of occurrences of each element of $S_\mathcal{B}$ in the multiset. If there is a bijection $g : S_\mathcal{B} \to S_\mathcal{X}$ such that $m_\mathcal{B} = m_\mathcal{X} \circ g$, then we say that $g$ is also a bijection between the multisets $\mathcal{B}$ and $\mathcal{X}$.

We note that $S_\mathcal{X} = \text{Im}[c]$ denotes the set of distinct colors in $G$. Without loss of generality, since we are interested in filtrations induced by $f$ on $G$, we can constrain ourselves to filter values on $S_\mathcal{X}$. Thus, filtrations induced by $f : S_\mathcal{X} \to \mathbb{R}$ are increasing (i.e., for any consecutive filtration steps $j > i$, we have that $V_j \setminus V_i \neq \emptyset$) and produce filtration steps $\mathcal{T} = \{f(x) \mid x \in S_\mathcal{X}\}$. Because such filtrations are increasing, we have at least one vertex discovered at each step, resulting in the set of distinct birth times $S_\mathcal{B} = \mathcal{T}$. The mapping $g : S_\mathcal{X} \to S_\mathcal{B}$ where $g(x) = f(x)$ for all $x \in S_\mathcal{X}$ is a bijection. By definition, the number of vertices discovered at step $f(x)$ equals the number of persistence pairs with birth time $f(x)$, which is also equal to the number of vertices of color $x$. This implies that the multiplicity of an element $x$ in $\mathcal{X}$ is the same as its corresponding element $g(x)$ in $\mathcal{B}$. $\square$

### B.6 Proof of Theorem 1: The expressive power of vertex-color filtrations

*Proof.* We consider graphs $G = (V, E, c, X)$ and $G' = (V', E', c', X')$. and adopt the following notation. We use $\mathcal{X} = \{\!\{c(v) \mid v \in V\}\!\}$ and $\mathcal{X}' = \{\!\{c'(v) \mid v \in V'\}\!\}$ to denote the multisets of vertex colors of $G$ and $G'$. Also, we denote by $C_1, \ldots, C_k$ the components of $G$, and by $C'_1, \ldots, C'_{k'}$ the components of $G'$. The set $X_i = \{c(w) \mid w \in V_{C_i}\}$ denotes the distinct colors appearing in $C_i$. Similarly, $X'_i = \{c'(w) \mid w \in V_{C'_i}\}$ refers to the distinct colors in $C'_i$.

[Forward direction $\Rightarrow$] $\mathcal{D}_G^0 \neq \mathcal{D}_{G'}^0 \to$ there is a color-separating set

The persistence diagrams $\mathcal{D}_G^0$ and $\mathcal{D}_{G'}^0$ for graphs with $\mathcal{X} = \mathcal{X}'$ have the same birth times. It implies that if both the real holes and almost holes are identical, then the diagrams are also identical. As such,

the assumption that $\mathcal{D}_G^0 \neq \mathcal{D}_{G'}^0$ gives that either (1) their multisets of real holes or (2) their multiset of almost holes are different. In the following, we consider these two cases.

Regarding case (1), Lemma 2 gives that if $\mathcal{D}_G^0 \neq \mathcal{D}_{G'}^0$ with different multisets of real holes, then we have that $\{\!\{X_i\}\!\}_{i=1}^k \neq \{\!\{X_i'\}\!\}_{i=1}^k$. Whenever this happens, the forward direction holds as even an empty set would work as a color-separating set here. Thus, it suffices to consider the case when $\mathcal{D}_G^0$ and $\mathcal{D}_{G'}^0$ only differ in their multisets of almost holes. In this case, we can directly leverage Lemma 4 to obtain that there is a color-separating set for $G$ and $G'$.

[Backward direction $\Leftarrow$] Now we want to show that if there is a color-separating set $Q \neq \emptyset$ for $G$ and $G'$, there exists a filtration such that $\mathcal{D}_G^0 \neq \mathcal{D}_{G'}^0$.

If $Q = \emptyset$, i.e. $G$ and $G'$ have distinct component-wise colors, the claim follows by Lemma 2, with $\mathcal{D}_G^0$ and $\mathcal{D}_{G'}^0$ having different multisets of real holes. If $Q \neq \emptyset$, we can however use Lemma 2 to subgraphs $G_{\bar{V}}$ and $G_{\bar{V}'}'$, induced by $\bar{V} = V \setminus \{w \in V \mid c(w) \in Q\}$ and $\bar{V}' = V' \setminus \{w \in V' \mid c'(w) \in Q\}$ to gain a filter function $g$ such that the diagrams for these subgraphs differ. Now, let's choose any a filter function such that $f(x) = g(x) \: \forall x \in X \setminus Q$ AND the filtration values for vertices with colors in $X \setminus Q$ are smaller than those with colors in $Q$. It follows there is a filtration step $j$ such that $G_j = G_{\bar{V}}$ and $G_j' = G_{\bar{V}'}'$, and that the birth times for real holes (if the vertices of colors in $Q$ do not merge the real holes of the subgraphs) or almost holes (if the vertices of colors in $Q$ do merge components that would have been real holes in the subgraphs) differ. Thus, $\mathcal{D}_G^0 \neq \mathcal{D}_{G'}^0$. $\qquad\square$

## B.7 Proof of Lemma 6: Edge-based almost holes as disconnecting sets

*Proof.* Initially, when none of the edges are added, there must be $|V|$ connected components. By definition, each pair $(0, d) \in \mathcal{D}^0$ corresponds to the death of one component. It follows that $G_{f(x^{(d)})}$ has $c = |V| - |\{(0, d) \in \mathcal{D}^0 \mid d \leq f(x^{(d)})\}|$ connected components. If $G$ has $\beta^0$ connected components, the subgraph with vertices $V$ and edges $E \setminus \{e \in E \mid f(l(e)) \geq f(x^{(d)})\}$ has more than $\beta^0$ connected components. Thus, $\{e \in E \mid f(l(e)) \geq f(x^{(d)})\}$ is a disconnecting set of $G$. $\quad\square$

## B.8 Lemma 7: The reconstruction of a disconnecting set

*Proof.* Let $\pi$ be the permutation of colors associated with a vertex-color filter function $f$, i.e., for a set of colors $X = (x_1, \ldots, x_m)$, we have that $f(x_{\pi(i)}) < f(x_{\pi(i+1)}) \: \forall \: i = 1, \ldots, m - 1$. Also, assume the colors associated with a disconnecting set $S$ of a graph $G = (V, E, l, X)$ is $X_S = \{x_{\pi(k)}, x_{\pi(k+1)}, \ldots, x_{\pi(m)}\}$.

If $S$ is a minimal disconnecting set, $(0, f(x_{\pi(k)}))$ must be an almost hole in $\mathcal{D}$ since if we could add edges $W \subseteq S$ with color $x_{\pi(k)}$ without killing some connected component of $G_{f(\pi(k-1))}$, the set of edges $S' = S \setminus W$ would form a proper disconnecting subset of a minimal disconnecting set If $S$ is not a minimal disconnecting, then there must be a proper subset $S' \subset S$ that is a minimal disconnecting set of $G$. Now, we choose $S'$ to be included first in the filtration, followed by elements of $S \setminus S'$. Thus, in both cases, there is a filtration s.t. an almost-hole $(0, f(x_k))$ appears, which allows us to reconstruct $S$. $\qquad\square$

## B.9 Proof of Theorem 2: The expressive power of edge-color filtrations

[$\Leftarrow$] We split the proof of the backward direction into three cases.

*Proof.* **Case 1: The color-disconnecting set $Q$ equals to $X \cup X'$.** This is a trivial case. If $Q = X \cup X'$, $G$ and $G'$ have distinct number of connected components when *all* the edges are removed from both graphs. This means $|V| \neq |V'|$. Now, if $|V| \neq |V'|$, then $|\mathcal{D}_G^0| \neq |\mathcal{D}_{G'}^0|$ for any filtration.

**Case 2: The color-disconnecting set $Q$ is an empty set.** Now, the graphs have distinct number of connected component (even if none of the edges are removed), i.e. $\beta_G^0 \neq \beta_{G'}^0$. The diagrams differ for any filtration since they have different numbers of real holes.

**Case 3: The color-disconnecting set $Q \neq \emptyset$ is a proper subset of $X \cup X'$:** The existence of a color-disconnecting set implies there is a set $S \subset X \cup X'$ such that by removing the edges of colors $S$, the two graphs will have different number of connected components. Without loss of generality, we can assume that after removing the edges of colors $S$, $G$ has more components than $G'$. Now,

we note that in a filtration where the colors of $S$ are added the latest, there must either be more almost holes $(0, f(x^{(d)}))$ in $\mathcal{D}_G^0$ than in $\mathcal{D}_{G'}^0$ with $x^{(d)} \in S$, or alternatively $\beta_G^0 \neq \beta_{G'}^0$. In both cases, $\mathcal{D}_G^0 \neq \mathcal{D}_{G'}^0$ for some filter function $f$. $\qquad\square$

[$\Rightarrow$] To prove the forward direction of the Theorem, we consider the cases where the edge-color diagrams differ in 1) their size, 2) the number of real holes, and 3) their almost holes.

*Proof.* **Case 1:** $|\mathcal{D}_G| \neq |\mathcal{D}_{G'}|$. Again, this corresponds to a trivial case, since if $|\mathcal{D}_G| \neq |\mathcal{D}_{G'}|$, then $|V| \neq |V'|$. Now, $Q = X \cup X'$ is a color-disconnecting set.

**Case 2:** $\mathcal{D}_G$ **and** $\mathcal{D}_{G'}$ **differ in their real holes**. If there is a different count of real holes, then $\beta_G^0 \neq \beta_{G'}^0$, and $Q = \emptyset$ is a color-disconnecting set.

**Case 3:** $\mathcal{D}_G$ **and** $\mathcal{D}_{G'}$ **only differ in their almost holes**. We now assume that $|\mathcal{D}_G| = |\mathcal{D}_{G'}|$ and $\beta_G^0 = \beta_{G'}^0$, but $\mathcal{D}_G \neq \mathcal{D}_{G'}$. This means that there is some $(0, d) \in \mathcal{D}$ such that there are more almost holes with this death time in $\mathcal{D}_G$ than in $\mathcal{D}_{G'}$, without loss of generality. There may be several such almost holes (for which the diagrams differ) with distinct death times. Let's denote the set of the death times for these almost holes by $D$. Then, let $d_{min}$ be the minimum of the death times in $D$, i.e. $d_{min} = \min_{d \in D} d$. Let us show that the set $Q = \{x \in X \cup X' \mid f(x) > d_{min}\}$ disconnects $G'$ into more connected components than $G$. For any lower filtration step, the induced subgraphs must have as many connected components because the almost holes corresponding to those steps match, and $|\mathcal{D}_G| = |\mathcal{D}_{G'}|$, i.e. $|V| = |V'|$, which means that at filtration step 0, we begin with equally many connected components. At filtration step $d_{min}$, we connect more components in $G$ than in $G'$ because there are more almost holes corresponding to this step in $\mathcal{D}_G$ than in $\mathcal{D}_{G'}$. Now, $Q$ must be a color-disconnecting set. $\qquad\square$

## B.10 Proof of Theorem 3: Edge-color vs. vertex-color filtrations

*Proof.* This Theorem is proved in Section 3.3. In particular, Figure 3(a) provides an example of pairs of graphs that can be distinguished by vertex-color filtrations but not from edge-color ones. On the other hand, the graphs in Figure 3(b) can be distinguished by edge-color filtrations but not from vertex-color ones. This concludes the proof. $\qquad\square$

## B.11 Proof of Theorem 4: RePHINE is isomorphism invariant

*Proof.* RePHINE diagram's isomorphism invariance stems from the fact that it is a function of a filtration on graph, and this filtration is gained from isomorphism invariant colorings. If this assumption is violated and the colorings are not gained in an invariant way, RePHINE diagrams can also be inconsistent.

It is easy to check that the tuples (b,d) are isomorphism invariant - when $b = 0$, these tuples correspond to diagrams gained from edge-color filtration. In this case, we can check the conditions given by Theorem 2 and note none of the conditions may be met with isomorphic graphs. With regard to $b = 1$, the set of missing holes is multiset of edge colors that did not appear in edge-color filtration diagram. This set can thus be gained by considering the multiset of edge colours and the edge-color diagram, which are both isomorphism invariant.

Further, it is also easy to see that the tuples $(\alpha, \gamma)$ are invariant. When $b = 0$, the set of $\alpha$'s corresponds to the multiset of vertex colours, and for each vertex, $\gamma = \min_{v \in \mathcal{N}(w)} f_e(\{c(w), c(v)\})$.

However, the crucial part is how these two tuples are concatenated, i.e., how each of the vertices are associated with real and almost-holes. In particular, we need to check when two connected components are merged at a filtration step $i$ and RePHINE compares the representatives (i.e. vertices of a connected component which have not yet 'died') of the two components, we will end up with same diagram elements of form $(b, i, \alpha, \gamma)$ regardless of the order we add the edges of color with filtration value $i$. In other words, while the RePHINE algorithm considers one edge at a time and does only pairwise comparisons between merged connected components, the order or these comparisons must not affect the decision on which vertices are associated with death time $i$. Let's consider what happens when adding all the edges of a color results in merging more than two components. Assume there is a new connected components constituting of old connected components $T_1, T_2, ..., T_n$. Now, there are two different cases. Assume first that there is a strict minimum among the vertex filtration values of the old representatives. Then, any pairwise comparison will lead to choosing this minimum

as the representative of the new connected component and all the other vertices will die at this filtration step. Then, assume there is no strict minimum but a tie between two or more representatives. Then, there will be comparisons based on $\gamma$, but choosing maximum of these is also permutation invariant function. In case there are two (or more) representatives such that there is a tie based on the vertex filtration values and $\gamma$ values, choosing at random any of these leads to the same diagram. Lastly, note that for each real hole, $(b, d) = (0, \infty)$, and so it does not thus matter how each of the vertices are matched to the real holes, when rest of vertices are associated with almost-holes in an invariant way. □

## B.12 Proof of Theorem 5: RePHINE is strictly more expressive than color-based PH

Let $\mathcal{R}_G$ denote the RePHINE diagram for a graph $G$. Similarly, let $\mathcal{D}_{v,G}$ and $\mathcal{D}_{e,G}$ denote persistence diagrams associated with vertex- and edge-color filtrations of $G$. We assume that $\mathcal{D}_{v,G} = (\mathcal{D}_{v,G}^0, \mathcal{D}_{v,G}^1)$ and $\mathcal{D}_{e,G} = (\mathcal{D}_{e,G}^0, \mathcal{D}_{e,G}^1)$ include 0- and 1-dim persistence diagrams. We want to show that for two graphs $G$ and $G'$

    (i) if there is a vertex-color filtration such that $\mathcal{D}_{v,G} \neq \mathcal{D}_{v,G'}$ then there is a filtration that lead to $\mathcal{R}_G \neq \mathcal{R}_{G'}$.

    (ii) if there is a edge-color filtration such that $\mathcal{D}_{e,G} \neq \mathcal{D}_{e,G'}$ then there is a filtration that lead to $\mathcal{R}_G \neq \mathcal{R}_{G'}$.

These results would show that RePHINE is at least as expressive as color-based persistence diagrams. We further show that

    (iii) there is a pair of non-isomorphic graphs for which we can obtain $\mathcal{R}_G \neq \mathcal{R}_{G'}$ but $\mathcal{D}_{v,G} = \mathcal{D}_{v,G'}$ and $\mathcal{D}_{e,G} = \mathcal{D}_{e,G'}$ for all vertex- and edge-color filtrations.

*Proof.* **Part (i)**: $\mathcal{D}_{v,G} \neq \mathcal{D}_{v,G'} \to \mathcal{R}_G \neq \mathcal{R}_{G'}$. Let $f$ be the vertex-color function associated with the standard diagrams $\mathcal{D}$. We can choose the RePHINE's vertex-level function $f_v$ such that $f_v = f$. We note that the original diagrams can be obtained from an auxiliary edge-level filter function $f_a$ where $f_a(u, w) = \max(f(c(u)), f(c(w)))$. The procedure is described in Algorithm 1.

Let $f_e$ be the edge-color filter function of RePHINE. If we choose $f_e = f_a$, then RePHINE contains in the second and third elements of its tuples exactly the same persistence information of the vertex-color diagrams. Note that in this case, we do not even need to require injectivity of the edge-color filter $f_e$ since the max function is not injective. Regarding the 1-dim features, for any tuple $(d, \infty)$ in the 1-dim persistence diagram, we have a missing hole $(1, d, \cdot, \cdot)$ that comprises the same information. Thus, we have constructed vertex- and edge-color functions such that $\mathcal{D}_{v,G} \neq \mathcal{D}_{v,G'} \to \mathcal{R}_G \neq \mathcal{R}_{G'}$.

**Part (ii)**: $\mathcal{D}_{e,G} \neq \mathcal{D}_{e,G'} \to \mathcal{R}_G \neq \mathcal{R}_{G'}$. This is a trivial case as RePHINE consists of an augmented version of standard edge-color diagrams. Let $f$ be the edge-color (injective) filter function associated with the standard diagrams. In this case, we can simply set $f_e = f$, where $f_e$ is RePHINE's edge-color filter. Then, the first and second elements of RePHINE's tuples correspond to $\mathcal{D}_e$. Regarding the 1-dim features, the only difference is the way the information is encoded. While we adopted the convention $(1, d)$ for missing holes, the standard diagrams often use $(d, \infty)$. The relevant information is the same. Therefore, RePHINE is at least as expressive as edge-color persistence diagrams.

**Part (iii)**. To show that RePHINE is strictly more expressive than color-based PH, it suffices to provide an example of two graphs for which there is a filtration such that $\mathcal{R}_G \neq \mathcal{R}_{G'}$ but these graphs cannot be separated from any vertex- or edge-color filtration. We use the pair of graphs in Figure 3(c). We note that these graphs have no cycles, making 1-dim persistence information trivial.

We first note that their multisets of colors are identical and there is no color-separating sets for these two graphs — i.e., there is no subset of colors whose removal would separate the graphs into distinct component-wise colors. Thus, by Theorem 1, there is no vertex-color filtration s.t. $\mathcal{D}_{v,G} \neq \mathcal{D}_{v,G'}$.

Also, we have that $|V| = |V'|$ and $X = X'$ and $\beta_G^0 = \beta_{G'}^0$, and there is no color-disconnecting set for $G$ and $G'$ (i.e., there is no edge colors whose removal would generate subgraphs with different number of components). By Theorem 2, these graphs cannot be separated by any edge-color filtration.

However, if we choose the filter functions $f_v(\text{'blue'}) = 1$, $f_v(\text{'orange'}) = 2$, $f_e(\text{'blue-blue'}) = 4$, and $f_e(\text{'blue-orange'}) = 3$, we obtain distinct RePHINE di-

agrams given by $\mathcal{R}_G = \{\!\{(0,4,1,4),(0,\infty,1,3),(0,3,2,3),(0,3,2,3)\}\!\}$ and $\mathcal{R}_{G'} = \{\!\{(0,\infty,1,3),(0,4,1,3),(0,3,2,3),(0,3,2,3)\}\!\}$. $\qquad\square$

## C Implementation details

### C.1 Datasets

Table S1 reports summary statistics of the real-world datasets used in this paper. For the IMDB-B dataset, we use uninformative features (vector of ones) for all nodes. NCI1, NCI109, Proteins, and IMDB-B are part of the TU Datasets[2], a vast collection of datasets commonly used for evaluating graph kernel methods and GNNs. MOLHIV is the largest dataset (over 41K graphs) and is part of the Open Graph Benchmark[3]. We also consider a regression task using the ZINC dataset — a subset of the popular ZINC-250K chemical compounds [19], which is particularly suitable for molecular property prediction [7].

Table S1: Statistics of the datasets.

| Dataset | #graphs | #classes | #node labels | Avg #nodes | Avg #edges |
|---|---|---|---|---|---|
| NCI1 | 4110 | 2 | 37 | 29.87 | 32.30 |
| IMDB-B | 1000 | 2 | - | 19.77 | 96.53 |
| PROTEINS (full) | 1113 | 2 | 3 | 39.06 | 72.82 |
| NCI109 | 4127 | 2 | 38 | 29.68 | 32.13 |
| MOLHIV | 41127 | 2 | 9 | 25.5 | 27.5 |
| ZINC | 12000 | - | 28 | 23.16 | 49.83 |

The cubic datasets (`Cubic08`, `Cubic10`, and `Cubic12`) comprise non-isomorphic 3-regular graphs with 8, 10, and 12 vertices, respectively. These datasets contain 5 (`Cubic08`), 19 (`Cubic10`), and 85 (`Cubic12`) graphs and can be downloaded at https://houseofgraphs.org/meta-directory/cubic. For each dataset, we create a balanced graph classification problem by randomly assigning each graph a binary class. Also, since the graphs do not have node features, we add a scalar feature to each vertex, i.e., $c(v) = 1$ for all $v$. However, this would make 1WL-GNNs and PH fail to distinguish any pair of graphs. Thus, we change the features of some arbitrary vertices of each graph, making $c(v) = -1$ for 1 vertex in graphs from `Cubic08`, 2 vertices in `Cubic10`, and 3 vertices in `Cubic12` — we denote the resulting datasets as `Cubic08-1`, `Cubic10-2`, and `Cubic12-3`. Given the modified datasets, we aim to assess if the existing methods can overfit (correctly classify all) the samples.

### C.2 Models

We implement all models using the PyTorch Geometric Library [10].

**Synthetic data.** The GNN architecture consists of a GCN with 2 convolutional layers followed by a sum readout layer and an MLP (one hidden layer) with ReLU activation. The resulting architecture is: `Conv(1, 36)` $\to$ `Conv(36, 16)` $\to$ `sum-readout` $\to$ `BN(16)` $\to$ `MLP(16, 24, 1)`, where BN denotes a batch norm layer [18]. For the PH model, we consider standard vertex-color filtration functions. In particular, we apply the same procedure as Hofer et al. [13], Horn et al. [15] to compute the persistence tuples. We only consider 0-dim persistence diagrams. The filtration function consists of an MLP with 8 hidden units and ReLU activation followed by a component-wise sigmoid function: `Sigmoid(MLP(1, 8, 4))` — i.e., we use 4 filtration functions with shared parameters. Since we can associate persistence tuples with vertices, we concatenate the resulting diagrams to obtain a $|V| \times (4*2)$ matrix $[\mathcal{D}_1^0, \mathcal{D}_2^0, \mathcal{D}_3^0, \mathcal{D}_4^0]$, where $\mathcal{D}_i^0$ denotes the 0-dim diagram obtained using the $i$-th filtration function. This procedure was also employed by Horn et al. [15]. The obtained diagrams are processed using a DeepSet layer with mean aggregator and internal MLP function ($\Psi$) with 16 hidden and output units, `MLP(4 * 2, 16, 16)`. We then apply a linear layer on top of the aggregated features. The overall DeepSet architecture is: `MLP(4 * 2, 16, 16)` $\to$ `Mean Aggregator` $\to$ `Linear(16, 16)`. Finally, we obtain class predictions using BatchNorm followed by a single-hidden-layer MLP with 16 hidden units: `BN(16)` $\to$ `MLP(16, 16, 1)`.

RePHINE uses the same overall architecture as the PH model. The only differences are that i) RePHINE tuples are 3-dimensional (as opposed to 2-dimensional in PH), and ii) RePHINE addi-

[2]https://chrsmrrs.github.io/datasets/
[3]https://ogb.stanford.edu

tionally leverages an edge-level filtration function. Such a function follows the architecture of the vertex-level one, i.e., `Sigmoid(MLP(1, 8, 4))`. We note that RePHINE tuples are 3-dimensional instead of 4-dimensional because we removed their uninformative first component (equal to 0) since we only use 0-dim diagrams. In other words, we do not leverage missing holes.

Regarding the training, all models follow the same setting: we apply the Adam optimizer [21] for 2000 epochs with an initial learning rate of $10^{-4}$ that is decreased by half every 400 epochs. We use batches of sizes 5, 8, 32 for the cubic08, cubic10, and cubic12 datasets, respectively. All results are averaged over 5 independent runs (different seeds). For all models, we obtain the expressivity metric by computing the uniqueness of graph-level representations extracted before the final MLP, with a precision of 5 decimals. Importantly, these choices of hyperparameters ensure that all models have a similar number of learned parameters: 1177 (RePHINE), 1061 (PH), and 1129 (GCN).

**Real-world data.** For computing the standard vertex-color persistence diagrams, we use the code available by Horn et al. [15], which consists of a parallel implementation in `PyTorch` of the pseudocode in Algorithm 1. Moreover, we apply a multiple filtration scheme and concatenate the 0-dim persistence diagrams to form matrix representations — again similarly to the design in [15]. Then, we apply a DeepSet architecture of the form: `MLP(TupleSize * nFiltrations, OutDim, OutDim)` $\rightarrow$ `Mean Aggregator` $\rightarrow$ `Linear(OutDim, OutDim)`. We use MLPs to define vertex- and edge-level filtration functions. For the 1-dimensional persistence tuples (or missing holes), we first process the tuples from each filtration function using a shared `DeepSet` layer and then apply mean pooling to obtain graph-level representations — this avoids possibly breaking isomorphism invariance by concatenating 1-dimensional diagrams. We sum the 0- and 1-dim embeddings and send the resulting vector to an `MLP` head. The resulting topological embeddings are concatenated with last-layer GNN embeddings and fed into a final MLP classifier.

We carry out grid-search for model selection. More specifically, we consider a grid comprised of a combination of $\{2, 3\}$ GNN layers and $\{2, 4, 8\}$ filtration functions. We set the number of hidden units in the `DeepSet` and GNN layers to 64, and of the filtration functions to 16 — i.e., the vertex/edge-color filtration functions consist of a 2-layer MLP with 16 hidden units. For the largest datasets (ZINC and MOLHIV), we only use two GNN layers. The GNN node embeddings are combined using a global mean pooling layer. Importantly, for all datasets, we use the same architecture for RePHINE and color-based persistence diagrams.

For the TUDatasets, we obtain a random 80%/10%/10% (train/val/test) split, which is kept identical across five runs. The ZINC and MOLHIV datasets have public splits. All models are initialized with a learning rate of $10^{-3}$ that is halved if the validation loss does not improve over 10 epochs. We apply early stopping with patience equal to 40.

**Comparison to PersLay.** We followed the guidelines in the official code repository regarding the choice of hyper-parameters. In particular, PersLay applies fixed filtration functions obtained from heat kernel signatures of the graphs with different parameters, resulting in extended and ordinary diagrams for 0 and 1-dimensional topological features. For `RePHINE+Linear`, we carry out a simple model selection procedure using grid-search for the number of filtration functions ($\{4, 8\}$) and the number of hidden units ($\{16, 64\}$) in the `DeepSet` models.

**Hardware.** For all experiments, we use Tesla V100 GPU cards and consider a memory budget of 32GB of RAM.

### C.3 Computing RePHINE diagrams

Algorithm 2 describes the computation of RePHINE diagrams. The pseudocode has been written for clarity, not efficiency. The replacement for $\infty$ in real holes depends on the choice of edge and vertex filter functions. In all experiments, we employed the logistic function to the output of the feedforward networks (i.e., filtered values lie in $[0, 1]$) and used 1 to denote the death time of real holes.

## D  Additional experiments

Here, we complement the experiments on synthetic data, providing illustrations of the learned persistence diagrams and reporting results obtained when we combine 0- and 1-dimensional diagrams.

In Figure S3, we show the concatenation of the learned persistence diagrams at the end of the training procedure for RePHINE and PH (i.e., `standard` vertex-color filtrations). In these examples, the RePHINE diagrams are different while the PH ones are identical. We can observe this behavior by

**Algorithm 2** RePHINE

---

**Require:** $V, E, \text{eValues}, \text{vValues}$         ▷ Vertices, edges, and edge/vertex-color filter values
   $\text{uf} \leftarrow \textsc{UnionFind}(|V|)$
   $\text{pers0} \leftarrow \text{zeros}(|V|, 4)$                   ▷ Initialize the persistence tuples
   $\text{pers1} \leftarrow \text{zeros}(|E|, 4)$
   $\text{pers0}[:, 3] \leftarrow \text{vValues}$             ▷ Pre-set the 'birth' times of each node
   $\textsc{sIndices}, \textsc{sValues} \leftarrow \textsc{Sort}(\text{eValues})$
   **for** $e, \text{weight} \in \text{Pair}(\textsc{sIndices}, \textsc{sValues})$ **do** ▷ Pair is equivalent to the $\text{zip}$ function in Python
      $(v, w) \leftarrow e$
      **if** $\text{pers0}[v, 4] = 0$ **then**
         $\text{pers0}[v, 4] \leftarrow \text{weight}$         ▷ Save the first filtration step that a node is discovered
      **end if**
      **if** $\text{pers0}[w, 4] = 0$ **then**
         $\text{pers0}[w, 4] \leftarrow \text{weight}$
      **end if**
      $\text{younger} \leftarrow \text{uf.find}(v)$          ▷ younger denotes the component that will die
      $\text{older} \leftarrow \text{uf.find}(w)$
      **if** $\text{younger} = \text{older}$ **then**             ▷ A cycle was detected
         $\text{pers1}[e, 1] \leftarrow 1$
         $\text{pers1}[e, 2] \leftarrow \text{weight}$
         $\text{pers1}[e, [3,4]] \leftarrow \infty$
         **continue**
      **else**
         **if** $\text{vValues}[\text{younger}] = \text{vValues}[\text{older}]$ **then**
            **if** $\text{pers0}[\text{younger}, 4] < \text{pers0}[\text{older}, 4]$ **then**     ▷ Additional disambiguation step
               $\text{younger}, \text{older}, v, w \leftarrow \text{older}, \text{younger}, w, v$   ▷ Flip younger, older, and node ids
            **end if**
         **else if** $\text{vValues}[\text{younger}] < \text{vValues}[\text{older}]$ **then**
            $\text{younger}, \text{older}, v, w \leftarrow \text{older}, \text{younger}, w, v$
         **end if**
      **end if**
      $\text{pers0}[\text{younger}, 2] \leftarrow \text{weight}$
      $\text{uf.merge}(v, w)$                ▷ Merge two connected components
   **end for**
   **for** $r \in \text{uf.roots}()$ **do**
      $\text{pers0}[r, 2] \leftarrow \infty$
   **end for**
   $\mathcal{R} \leftarrow \textsc{Join}(\text{pers0}, \text{pers1})$
   **return** $\mathcal{R}$

---

carefully inspecting the multisets of vectors at each row of the concatenated tuples (each row of the plots in Figure S3). For instance, consider the diagrams in Figure S3(b): in the RePHINE diagram for the top graph, there is a row with 3 yellow entries which do not appear at the diagram for the bottom graph. However, the representations obtained from `Standard PH` are identical for these graphs.

In Figure 6 we reported results using only 0-dimensional topological features. For completeness, Figure S4 shows learning curves when exploiting both 0 and 1-dimensional diagrams. Overall, we can again observe that RePHINE produces higher expressivity and better fitting capability in comparison to vertex-color persistence diagrams.

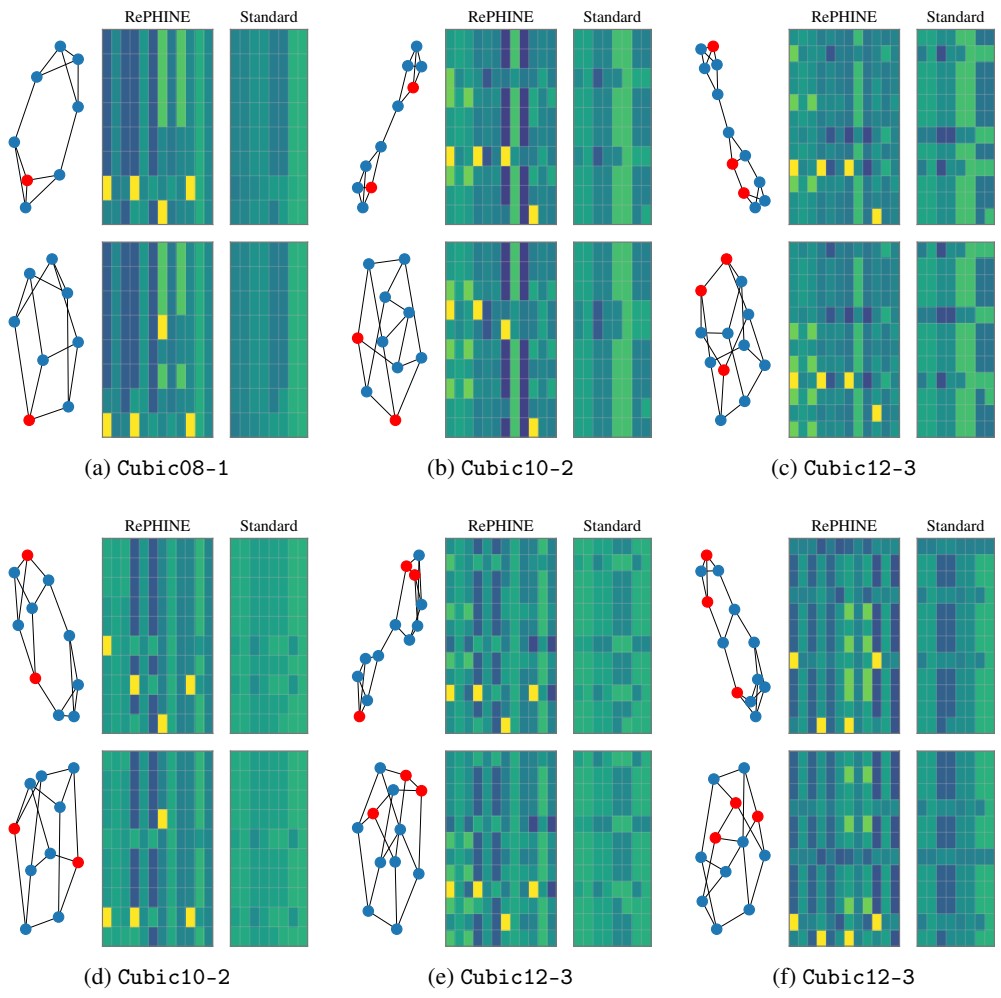

Figure S3: 0-dimensional diagrams obtained from RePHINE and PH (standard vertex-color filtrations). These represent pairs of graphs for which the learning procedure in RePHINE could yield different representations, whereas PH produced identical graph-level embeddings.

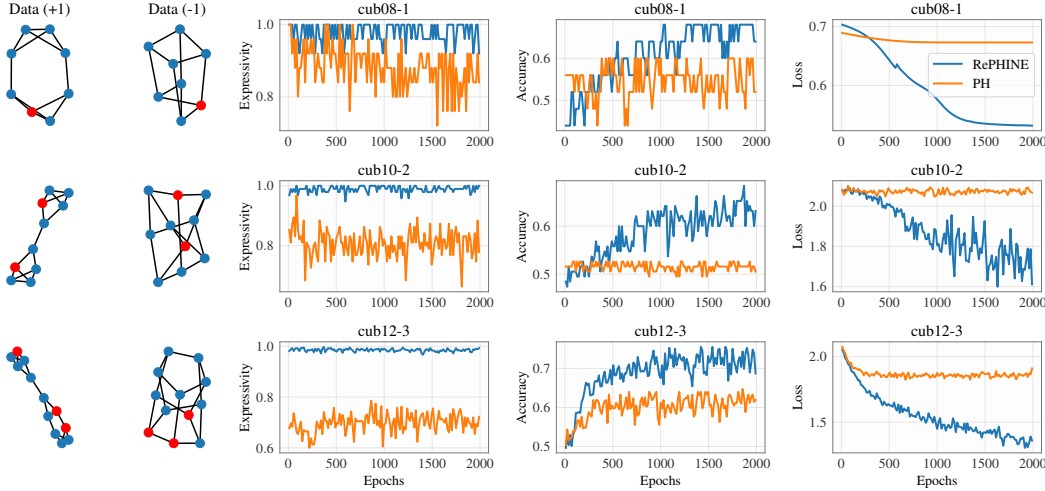

Figure S4: Average learning curves for RePHINE and PH on cubic graphs, using both 0-dim and 1-dim persistence diagrams. Again, RePHINE can better fit the graph samples.

