# OpenReview forum: "Going beyond persistent homology using persistent homology"
_NeurIPS.cc/2023/Conference — NeurIPS 2023 oral_

### Official Review · Reviewer_BFX1 · 2023-07-03

**Soundness:** 4 excellent
**Presentation:** 4 excellent
**Contribution:** 3 good
**Rating:** 7
**Confidence:** 5

**Summary:**

- The paper presents a comprehensive analysis of two types of color filtrations on graphs, focusing on their expressiveness.

- The paper introduces a novel topological summary RePHINE that combines both node and edge persistence diagrams. The proposed summary is proven to be more expressive than either node or edge persistence diagrams alone.

- The authors conduct experiments on synthetic and real-world datasets. They leverage a combination of RePHINE and a GNN structure to evaluate its performance.

**Strengths:**

- The presented method RePHINE for mixing 0- and 1- dim information is interesting and, as far as I know, is novel.
- A remarkable strength of the proposed method is its theoretical expressiveness, which surpasses the capabilities of utilizing 0- or 1-dimensional information in isolation.
- The paper is well presented and effectively communicates its ideas, ensuring a high level of clarity and ease of understanding for readers.

**Weaknesses:**

One weakness of the paper lies in the experimental evaluation section, which could benefit from a more comprehensive comparison with existing methods, such as the method proposed in [28]. Specifically, the paper could have included a comparative analysis between the proposed method and the local Persistent Homology (PH) method based on subgraph persistence diagrams included in  [28].

**Questions:**

- line 47-52: I think there is some syntax issue with the appearance of [Theory], [Methodology] and [Experiments]
- line 142: "disconnect" should be "disconnects".
- line 172: please mention that Q is a color separating set in this sentence.
- line 502: what does "disjoint union" here mean? What does "these sets" refer to? could you please provide an explicit description of the set you are constructing here?
- line 510: I don't think $\{\{X_i\}\}$ should have $l$ as a superscript. Are you also referring to connected components with $l$ colors instead of subgraphs?
- line 513: "l+1 the" -> "l+1 th"
- line 693: "it is not injective" should be "is not injective".

**Limitations:**

The limitations of the method are not adequately discussed. I think it would be beneficial for the authors to discuss applicability of the proposed method to handle large scale graphs.

There are no potential negative societal impact of the work.

---

> ### Author Rebuttal · Authors · 2023-08-09
>
> Thanks for your constructive feedback. In the following, we address all your questions.
>
> **W1: "One weakness of the paper lies in the experimental evaluation section, which could benefit from a more comprehensive comparison with existing methods, such as the method proposed in [28]."**
>
> We note that our initial experiments mainly focused on corroborating our theoretical analysis (synthetic datasets) and on assessing the benefits of combining RePHINE diagrams with popular GNNs for improved predictive performance. However, thanks to your comment, we have also compared RePHINE with a method that leverages extended persistence diagrams: PersLay. Our new results (please see Table 1 of the attached PDF) demonstrate that RePHINE outperforms PersLay on popular graph classification benchmarks.
>
> We have opted for PersLay instead of [28] due to the similarity between the evaluation setups and the ease of adaptation --- [28] mainly considers node classification tasks. Moreover, we are currently running experiments to investigate further benefits of integrating the RePHINE diagrams into modern GNNs (e.g., PNA, SpecFormer). We will include these results in the revised version of the manuscript.
>
> **Q1: "I think there is some syntax issue with the appearance of [Theory], [Methodology] and [Experiments]"**
>
> Thanks for pointing this out. We will make sure that there will be no such syntax issues in the revised manuscript.
>
> **Q2: "disconnect" should be "disconnects"**.
>
> Yes, thanks for the careful reading.
>
> **Q3: please mention that Q is a color separating set in this sentence.**
>
> We have modified that in the revised paper. Now the sentence reads:
>
> *We note that when $G$ and $G'$ have identical component-wise colors, the sets $\\{w \in V | c(w) \in Q\\}$ and $\\{w \in V' | c'(w) \in Q\\}$ induced by the color-separating set $Q$ are separating sets for $G$ and $G'$, respectively.*
>
> **Q4: "what does "disjoint union" here mean? What does "these sets" refer to? could you please provide an explicit description of the set you are constructing here?"**
>
> We agree that the construction was not clear. Thanks for pointing this out.  We meant that when $G$ and $G'$ have distinct component-wise colors, there must be some connected component $C_h$ in $G$ such that $\\{X_h\\} \neq \\{X'_j\\} $ for $ \forall 1 \leq j \leq k $.
>
> Then, if assumed $\beta^0_G = \beta^0_{G'}$, the unmatched component-colors come in pairs i.e. $G$ must have as many unmatched component-color sets as $G'$.  The collection of sets of unmatched component-color pairs is considered in the proof.
>
> We have clarified the language throughout the proof, and rewritten the mentioned part as follows :
>
> *When $G$ and $G'$ have distinct component-wise colors [math was ommited here due to markdown issues], there must be at least one connected component $C_h$ in $G$ such that $\\{X_h\\} \neq \\{X'_j\\} $ for $ \forall 1 \leq j \leq k $. Let us now call such $\\{X_h\\}$ a set of unmatched component colors*.
>
> **Q5: "I don't think $X_i$ should have $l$ as a superscript. Are you also referring to connected components with $l$
>  colors instead of subgraphs?"**
>
> This is a typo, the superscript should be $k$. We fixed it.
>
> **Q6: line 513: "l+1 the" -> "l+1 th"**.
>
> We have fixed that typo in the revised paper.
>
> **Q7: "it is not injective" should be "is not injective".**
>
> Yes, thanks for the careful reading.
>
> ---
> We hope our answers have addressed the points you have raised, and improved your view of the manuscript.

---

> > ### Comment · Reviewer_BFX1 · 2023-08-16
> >
> > Thank you for addressing my questions.
> >
> > Upon reading your response to Q4, I have some further questions:
> >
> > The authors highlighted that "when $G$ and $G'$ have distinct component-wise colors, there must be some connected component $C_h$ in $G$ such that $\{X_h\} \neq \{X_j'\}$ for all $1 \leq j \leq k$." My understanding is that when referring to the distinct component-wise colors of the two graphs, we're essentially discussing the differences between two multisets. However, two distinct multisets can still share the same individual elements. Given this, I'm not entirely convinced about the validity of the aforementioned claim.
> >
> > Could the authors provide further explanation?

---

> > > ### Author Response · Authors · 2023-08-16
> > >
> > > We apologize for the confusion. Our idea was to convey that when these multisets differ, there cannot be a bijection between the connected components of G and G' such that each component would be paired with a component having the same component-wise colors.
> > >
> > >
> > > Thanks to your comment, we have reformulated this part of the proof in a way that we believe is much more accessible for the readers. Thus, the base case in Lemma 2 now reads:
> > >
> > >
> > >
> > > ---
> > >
> > > If there is only one color, component-wise colors cannot differ for graphs with $\beta^0_G = \beta^0_{G'}$. Let us consider two colors (say, $b$ and $w$). For two colors, there are only three possibilities for what $X_h$
> > > in $\\{\\{X_i\\}\\}_{i=1}^k$
> > > may be:
> > >
> > > $\\{ b \\},\\{ w \\}$ or $\\{b, w \\} $.
> > >
> > >
> > >
> > > Now, let us denote the multiplicities of $\\{ b \\},\\{ w \\}$ and $\\{b, w \\} $ in $\\{\\{X_i\\}\\}_{i=1}^k$ by $n_1, n_2$ and $n_3$, respectively.
> > >
> > >
> > >
> > > For $G$ and $G'$ with $\beta^0_G = \beta^0_{G'}$, we have that $n_1 + n_2 + n_3 = n'_1 + n'_2 + n'_3$.
> > >
> > >
> > >
> > > Thus, when $\\{\\{X_i\\}\\}_{i=1}^k \neq$
> > >
> > >  $\\{\\{X_i^\prime\\}\\}_{i=1}^k$, there are four cases to consider:
> > >
> > >
> > >
> > > 1. $n_1 \neq n'_1, n_2 \neq n'_2,n_3 = n'_3 $: In this case, $n_2 + n_3 \neq n'_2 + n'_3 $ correspond to multiplicities of real holes $(w,\infty)$ for $G$ and $G'$ respectively, in a filtration that introduces the color $w$ first.
> > >
> > > 2. $n_1 \neq n'_1, n_2 = n'_2,n_3 \neq n'_3 $ : Again, $n_2 + n_3 \neq n'_2 + n'_3 $ correspond to multiplicities of real holes $(w,\infty)$ for $G$ and $G'$ respectively in a filtration that introduces the color $w$ first.
> > >
> > > 3. $n_1 = n'_1, n_2 \neq n'_2,n_3 \neq n'_3 $: Now, $n_1 + n_3 \neq n'_1 + n'_3 $ correspond to multiplicities of real holes $(b,\infty)$ for $G$ and $G'$ respectively in a filtration that introduces the color $b$ first.
> > >
> > > 4. $n_1 \neq n'_1, n_2 \neq n'_2,n_3 \neq n'_3 $: Similarly, $n_1 + n_3 \neq n'_1 + n'_3 $ correspond to multiplicities of real holes $(b,\infty)$ for $G$ and $G'$ respectively in a filtration that introduces the color $b$ first.
> > >
> > >
> > >
> > > Note that cases as $n_1 \neq n'_1, n_2 = n'_2,n_3 = n'_3 $ are not possible as $n_1 + n_2 + n_3 = n'_1 + n'_2 + n'_3$.
> > >
> > >
> > >
> > > ---
> > >
> > > Thanks again for checking the proofs. We genuinely appreciate your contribution to strengthening the paper.

---

> > > > ### Comment · Reviewer_BFX1 · 2023-08-19
> > > >
> > > > Thank you for the clarification and I will maintain my supportive score of the paper.

---

### Official Review · Reviewer_oULm · 2023-07-03

**Soundness:** 2 fair
**Presentation:** 3 good
**Contribution:** 2 fair
**Rating:** 5
**Confidence:** 3

**Summary:**

In this paper, the authors discuss the limitations of message-passing graph neural networks (MP-GNNs) in terms of the Weisfeiler-Leman test for isomorphism. They explore the use of persistent homology (PH) to augment graph models with topological features but highlight the challenge of identifying the class of attributed graphs that PH can recognize. To address this problem, they introduce the concept of color-separating sets. They establish necessary and sufficient conditions for distinguishing graphs based on the persistence of their connected components using filter functions on vertex and edge colors. Based on these insights, they propose RePHINE, a method for learning topological features on graphs. RePHINE integrates both vertex-level and edge-level PH, claimed to be more powerful than either category alone. When incorporated into MP-GNNs, RePHINE enhances their expressive power.

**Strengths:**

The paper presents new theoretical results and introduces a concept of color-separating sets, providing a resolution to the problem of recognizing attributed graphs based on the persistence of their connected components. The authors establish necessary and sufficient conditions for distinguishing graphs using filter functions on vertex and edge colors. They also propose RePHINE, a method for learning topological features on graphs, which integrates both vertex-level and edge-level persistent homology.

**Weaknesses:**

While the theoretical contributions of this paper are new, there is room for further exploration and validation in real-world experiments. The current evaluation primarily focuses on controlled simulated datasets, limiting our understanding of RePHINE's performance in practical scenarios. It is equally important to conduct more comprehensive experiments using real-world data to fully assess the efficacy and applicability of the proposed approach. Addressing these limitations would strengthen the practical relevance of the paper's findings. Are there any specific limitations or challenges when applying RePHINE to real-world applications? Additionally, it would be interesting to understand the factors that contribute to the marginal performance gain of RePHINE on most real-world datasets.

**Questions:**

Please refer to the sections above.

**Limitations:**

Please refer to the sections above.

---

> ### Author Rebuttal · Authors · 2023-08-09
>
> Thanks for your feedback. We reply to your comments/questions below.
>
> **"There is room for further exploration and validation in real-world experiments. The current evaluation primarily focuses on controlled simulated datasets, limiting our understanding of RePHINE's performance in practical scenarios."**
>
> While we agree that there is room for improvements, we would like to highlight that our initials experiments were mainly designed to 1) validate our theoretical findings (synthetic datasets), and 2) show that the proposed diagrams can be easily integrated into GNNs to boost their predictive performance on popular benchmarks on graph classification. Thanks to your comment, we are currently running experiments to investigate further benefits of integrating the RePHINE diagrams into modern GNNs (e.g., PNA, SpecFormer). We will include these results in the revised version of the manuscript.
>
> Importantly, we have now compared RePHINE diagrams with Extended Persistence diagrams used in the PersLay model. Our results demonstrate that RePHINE outperforms PersLay on four real-world datasets by a large margin (please see Table 1 in the attached pdf). Thus, we believe that beyond the theoretical analyses, the empirical benefits of RePHINE can already be demonstrated.
>
> **Are there any specific limitations or challenges when applying RePHINE to real-world applications?**
>
> We believe that topological descriptors (such as RePHINE) work complementary to existing graph classifiers rather than standalone methods. As a standalone method, we know RePHINE has theoretical limitations. In particular, for unattributed graphs, RePHINE cannot separate graphs of equal size with the same number of components and cycles (please see the example in the attached PDF).
>
> However, given RePHINE's improved expressivity at the same computational cost as vanilla vertex-color filtrations, RePHINE has the potential to become the default choice for topologically enriched graph models in real-world applications.
>
> **It would be interesting to understand the factors that contribute to the marginal performance gain of RePHINE on most real-world datasets.**
>
> While it is hard to provide a definitive explanation for subtle performance differences, we believe that the minor gains come from our specific architectural choices, which isolate persistence diagrams while keeping other components unchanged. We note that in the comparison against the PersLay model, we can observe a significant difference in performance.

---

> ### Comment · Area_Chair_YCf2 · 2023-08-18
>
> Dear reviewer,
>
> Please **briefly acknowledge the rebuttal** by the authors and consider updating your score—we want to avoid borderline scores for reviews, and the discussion phase will close soon. If you have any additional questions to the authors  please ask them **now**.
>
> Thanks,\
> Your AC

---

> > ### Author Response · Authors · 2023-08-18
> >
> > Dear AC,
> >
> > Many thanks for your kind reminder.
> >
> >
> > Dear reviewer,
> >
> > Thank you again for your feedback. Given that the author-reviewer discussion deadline is approaching, we would like to highlight additional experiments on real-world benchmarks to alleviate your concerns.
> >
> > The first set of experiments compares RePHINE to another PH-based model: PersLay (AISTATS, 2020) (for details, please see x2he W1/Q1). The results are:
> > | Method    | NCI109 | PROTEINS | IMDB-B | NCI109 |
> > | -------- | ------- | ------- | ------- | ------- |
> > | PersLay  |  90.48 $\pm$ 2.97   | 94.64 $\pm$ 4.69 | 90.40 $\pm$ 4.90 | 85.16 $\pm$ 6.11 |
> > | RePHINE+Linear |   **93.97 $\pm$ 4.42**   | **98.93 $\pm$ 3.39** |  **94.70 $\pm$ 7.50** | **93.80 $\pm$ 4.05** |
> >
> > As we observe, RePHINE+Linear outperforms PersLay by a significant margin.
> >
> > We also ran experiments regarding the combination of RePHINE and a SOTA GNN: PNA (NeurIPS 2020). We report results on the ZINC dataset. The MAE values are: PNA (0.195 $\pm$ 0.004) and PNA+RePHINE (**0.189 $\pm$ 0.006**). Importantly, we did our best to conduct a fair comparison with reproducible results. We will include these and a few others in our revised manuscript.
> >
> > Finally, we report fundamental theoretical results to uncover the representational limits of PH. These, alongside a provably more expressive topological descriptor, are our main contributions. We expect our work will help the Graph ML and Topological DL communities design better, more nuanced models that are both theoretically well-grounded and practically efficacious.
> >
> > Thank you again for taking the time to review our submission and for your constructive feedback. We would greatly appreciate if you would kindly consider upgrading your score.

---

> > > ### Comment · Reviewer_oULm · 2023-08-20
> > >
> > > Thank you for the rebuttal. I hold a favorable view of the supplementary experiment aimed at addressing the initial concerns, even though the enhancement does not show significant statistical difference. I will maintain my original evaluation stance and defer the final judgment to the AC.

---

### Official Review · Reviewer_rXNh · 2023-07-06

**Soundness:** 4 excellent
**Presentation:** 4 excellent
**Contribution:** 4 excellent
**Rating:** 7
**Confidence:** 3

**Summary:**

The authors introduce RePHINE, which calculates 0-dimensional persistent homology (PH) with respect to the filtration on edge colors, augmented with so-called missing holes and vertex color information. They establish the necessary and sufficient conditions for distinguishing graphs. RePHINE is shown to be more expressive than both standard 0-dim and 1-dim PH, can be easily integrated into GNNs, and is demonstrated to boost their expressive power on several benchmarks for graph classification.

**Strengths:**

(S1) The three questions in the Introduction nicely position and motivate the work, and the presentation of the main contributions is very clear.

(S2) The theoretical results are relevant, as they discuss in detail the expressivity of PH on vertex-color and edge-color filtrations, and expressivity of RePHINE. I do not have a good overview of PH on graphs, so I cannot comment on the novelty of results, and I did not check the supplementary material for proofs.

(S3) Experiments are carried out on 3 synthetic and 5 real-world datasets.


**Weaknesses:**

(W1) The related work seems not be detailed enough and is hard to identify.

(W2) The name RePHINE is clever and nice. However, the word interleaving suggests interleaving distance (esp. relevant in TDA), so it would probably be better to replace this with e.g. interplay or integration. Also, it is somewhat a misnomer since you are not really using a filtration on nodes (this information does not influence birth and death values), so it might be better to rephrase that part too (e.g. Refined PH by incorporating node-color on edge-based filtration)?

(W3) From the beginning of the paper, I was confused what the particular interplay between vertex- and edge-based PH will be. This was most pronounced in Section 3.3, since your way of edge-coloring definition, if used together with vertex-coloring, does not satisfy the definition of simplicial complex. For example, f(orange)=1, f(blue-orange)=2, f(blue)=3, so that an edge can appear in the filtration before the two incident vertices appear. Stressing earlier on (already in Section 1) that you calculate PH on edge-based filtration, including so-called missing holes and augmenting it with vertex-color information would be helpful. See related comment (W2) on the acronym RePHINE above.

(W4) You write: “We note that missing holes correspond to cycles obtained from 1-dim persistence diagrams.” How does you approach compare to concatenated standard 0- and 1-dim PH on edge-based filtration?

(W5) Experimental results on synthetic data: More information about the data should be included (in Appendix C.1). What exactly is the problem/goal here? What type of graphs results in the same RePHINE representation (in particular for cub12-3)? You write that you compare with 0- and 1-dim PH on vertex-color filtration, but what exactly do you mean by this, is this information concatenated (union of sets is considered)? What about PH on edge-color filtration? Why are standard PH and GNNs performing so poorly?

(W6) Experimental results on real-world data: You write that you compare against standard color-based PH, but what do you mean with this? See related comments in (W4). It would be interesting to look at the results in more detail, in particular to discuss some examples that would be wrongly classified with other approaches, but that are successfully tackled with your method, but also at the examples that cannot be classified properly with your approach (the discussion may be placed in an appendix).


**Questions:**

Main questions are formulated in the weaknesses (W1)-(W6) above.

(Q1) Abstract: “… provably more powerful than both”. The word both is confusing here, is your method more powerful than the two combined, or more powerful than each of them separately?

(Q2) What is a suitable metric for the augmented persistence diagrams? A brief comment is sufficient, and can be placed in future research.

(Q3) Which software do you use for calculation of PH?

Some minor suggestions:

-	“Experiments support our theoretical analysis and show the effectiveness of RePHINE on five real-world datasets.” -> “Experiments support our theoretical analysis and show the effectiveness of RePHINE on three synthetic and five real-world datasets.”
-	Provide a reference for persistence diagram (e.g., in Section 2), so that readers can find more information, and to make it clear that this is not defined for the first time here in your work.
-	Explicitly mention that all lemmas and theorems are proved in Appendix B, since one might wonder if some or all of these are earlier results.
-	Lemma 1 Vertex-based filtrations … -> Lemma 1 Injective vertex-based filtrations …
-	For better readability, it would be good to also have explicit Definitions for separating and disconnecting set. Improve consistency in naming theoretical results, e.g., if Lemma 6 (Edge-based almost holes as disconnecting sets), then it is better that Lemma 4 (Vertex-based almost holes as color-separating sets).

-	A visual or table summary (can be placed in Appendix B) of your theoretical results could really help the readability of Section 3 and the impact of your work. For example, some of the table rows could be the following:
1) Real holes (d = infty) of 0-dim PH wrt vertex-color filtration --- Component-wise colors --- Lemma 2
2) Almost holes (b neq d, d < infty) of 0-dim PH wrt vertex-color filtration --- Color-separating sets --- Lemma 3
3) Birth time of 0-dim persistence interval wrt vertex-color filtration --- Vertex color --- Lemma 5
4) Almost holes  (b neq d, d < infty) of 0-dim PH wrt edge-color filtration ---- Disconnecting sets --- Lemma 6

-	What is {{ on line 135, line 160, line 182, line 199, …?
-	In Section 3, you denote birth and death values with b and d (I think this is more common and readable), but you use a and b in Section 4.
-	Introduce the augmented PH as an explicit Definition in Section 4, as this is your main contribution.
-	Cite specific Appendix (e.g. Appendix B), rather than pointing to the general Appendix.
-	Capitalization in References: PersLay, Kolmogorov, Rayleigh-Bénard, Leman


**Limitations:**

Limitations and future work are not discussed.

---

> ### Author Rebuttal · Authors · 2023-08-09
>
> Thank you for your detailed and thoughtful review. You have raised very pertinent points. Below, we address your questions/comments.
>
> **W1: "The related work seems not be detailed enough and is hard to identify."**
>
> Thanks for your comment. In the Introduction, we decided to group references together for conciseness. To alleviate the issue you raised, we will provide a more detailed overview of related works in the supplementary material (in a new section Related Works).
>
> **W2: "The name RePHINE is clever and nice. However, the word interleaving suggests interleaving distance (esp. relevant in TDA), so it would probably be better to replace this with e.g. interplay or integration."**
>
> Thanks very much for the excellent suggestion. We agree that 'Refined PH by incorporating node-color into edge-based filtration' is apt for the proposed method, so have accordingly decided to adopt this rephrasing.
>
> **W3: "Stressing earlier on (already in Section 1) that you calculate PH on edge-based filtration, including so-called missing holes and augmenting it with vertex-color information would be helpful."**
> We have reworded descriptions in the Abstract, Introduction, and Section 4 to better reflect how RePHINE works. For instance, the Abstract now reads: "RePHINE efficiently incorporates vertex-color information into edge-level filtrations, achieving a scheme ...".
>
> **W4: "How does you approach compare to concatenated standard 0- and 1-dim PH on edge-based filtration?"**
>
> We note that RePHINE is more expressive than the union of the multisets of 0- and 1-dim PH on edge-based filtration. In particular, the two graphs in Figure 4(c) of the paper cannot be distinguished by either 0-dim or 1-dim PH. However, we can obtain two different RePHINE diagrams for such a graph as we show in part (3) of the proof of Theorem 4.
>
> **W5: Regarding details and further analysis of the results on synthetic data.**
>
> The cubic datasets comprise non-isomorphic 3-regular graphs. From these graphs, we create a classification problem by assigning each graph a binary class. Given the partition, we assess if the existing methods can overfit (correctly classify all) the samples. When we compare RePHINE with 0- and 1-dim PH on vertex-color filtration, we mean the union of the 0- and 1-dim diagrams.
>
> Regarding the analysis, we have only compared RePHINE to vertex-color filtration as it has been used in the reference work (TOGL). Nonetheless, in the revised version of the paper, we will also include results for diagrams obtained from edge-level filtrations. To understand why PH works poorly, we report diagrams (after learning) for two instances of cubic-10-2 in the attached PDF. We noticed that the original code of TOGL for vertex-color filtrations uses $max(f(c))$ instead of infty, which makes it not distinguish almost and real holes in some instances. We will also provide a similar discussion on the reasons for the failure of GCNs and report obtained diagrams for other graphs (and synthetic datasets) in the supplementary material.
>
> **W6: "Experimental results on real-world data: You write that you compare against standard color-based PH, but what do you mean with this?"**
>
> By standard color-based PH, we mean 0-dim and 1-dim persistence diagrams obtained from vertex-color filtrations. To the best of our knowledge, edge-color filtrations have not been used in graph learning. We will clarify this in the revised manuscript.
>
> **Q1: "Is your method more powerful than the two combined, or more powerful than each of them separately?"**
>
> RePHINE is more expressive than the union of the families of node- and edge-color filtrations. In particular, in Theorem 4, saying that RePHINE is strictly more expressive than vertex- *or* edge-level filtration implies that it is more powerful than they combined (union only allows the separation of graphs that can be separated by one of the filtration types).
>
> **Q2: "What is a suitable metric for the augmented persistence diagrams?"**
>
> This is an interesting question. In particular, future research could consider the suitability of the bottleneck distance for RePHINE diagrams with necessary changes.
>
> **Q3: "Which software do you use for calculation of PH?"**
>
> Our code is based on the official repo of Topological GNNs. As such, we have modified their routine to compute our augmented diagrams. It consists of a Torch implementation using a Find-Union data structure.
>
> In the following, we list the minor suggestions. For conciseness, we mark with $\checkmark$ the accepted suggestions.
> 1. $\checkmark$ “Experiments ... five real-world datasets.” $\rightarrow$ “Experiments ... on three synthetic and five real-world datasets.”
> 2. $\checkmark$ Provide a reference for persistence diagram (e.g., in Section 2)
> 3. $\checkmark$ Explicitly mention that all lemmas and theorems are proved in Appendix B.
> 4. $\checkmark$ Lemma 1 Vertex-based filtrations … $\rightarrow$ Lemma 1 Injective vertex-based filtrations …
> 5. It would be good to also have explicit Definitions for separating and disconnecting set. --- **This might be adopted if the page limit allows.**
> 6. $\checkmark$ Improve consistency in naming theoretical results.
> 7. $\checkmark$ A visual or table summary (can be placed in Appendix B) of your theoretical results could really help the readability of Section 3 and the impact of your work.} **See attached PDF.**
> 8. $\checkmark$ \textbf{What is $\{\{$ on line 135, line 160, line 182, line 199?} **We use $\{\{$ to represent multisets.**
> 9. $\checkmark$ In Section 3, you denote birth and death values with b and d, but you use a and b in Section 4.} **We are now using $b$ and $d$, as suggested.**
> 10. $\checkmark$ Introduce the augmented PH as an explicit Definition in  Section 4.
> 11. $\checkmark$ Cite specific Appendix (e.g. Appendix B).
> 12. $\checkmark$ Capitalization in References
>
> ---
>  We're grateful for your thoughtful and perceptive comments, and hope our answers have improved your assessment of this work.

---

> > ### Comment · Reviewer_rXNh · 2023-08-14
> >
> > Thanks a lot for the detailed response and for the great improvements! Please make sure to include most of these clarifications in the final version of the paper.

---

> > > ### Author Response · Authors · 2023-08-16
> > >
> > > Sure. We will include all clarifications in the revised version of the paper. Thanks again for your insightful review and support of our work.

---

### Official Review · Reviewer_x2he · 2023-07-07

**Soundness:** 3 good
**Presentation:** 2 fair
**Contribution:** 2 fair
**Rating:** 6
**Confidence:** 3

**Summary:**

The discriminative power of the persistent homology (in certain homological degrees) of vertex- and edge-filtered graphs is characterized in terms of combinatorial structure of the graphs. It is shown that there exist pairs that can be distinguished by the persistent homology of vertex-filtrations but not by the persistent homology of edge-filtrations, and viceversa. This is used to motivate the introduction of a new topological descriptor of graphs which is strictly more discriminative than the persistent homology of both vertex- and edge-filtrations.
The performance of this descriptor is evaluated on benchmark datasets.

[Post rebuttal edit] Raised score to 6.

**Strengths:**

1. The theoretical results on the expressiveness of persistent homology of vertex- and edge-filtered graphs are interesting, and they are put to good use in that they motivate the design of a provably stronger topological descriptor of graphs.

2. The fact that the invariant being introduced is strictly more discriminative than the persistent homology of vertex- and edge-filtrations seems to be relevant in practice.

**Weaknesses:**

3. The exposition is, at times, not that easy to follow. See, in particular, my comment about line 174, below.

4. The experimental section is interesting, but the empirical claims about the performance of the RePHINE would be better substantiated with experimental evaluation on more datasets.

5. The conclusions are a bit too succinct. In particular, the limitations of the approach are not discussed in detail. Relatedly, no future work or open questions are mentioned.

**Questions:**

## Main questions

6. Your approach seems similar to extended persistence (in the sense of your reference [2]).
How does your main strategy (RePHINE) compare to extended persistence and in particular to Perslay (both in theory and in practice)?
Can you substantiate this with theory or experiments?

7. Is it clear that RePHINE is isomorphism invariant? Since this is an important property for the theorical section of this paper, I believe that this should be addressed explicitly.

8. Is RePHINE stable in the sense of persistent homology? Is this relevant in your setup?

9. In line 58 you mention that you aim to fully characterize graphs that can be distinguished with persistent homology.
Was this achieved or are there still open questions?

## Minor questions and comments

10. Line 61: Should the subset inclusion be $\in$?

11. Line 135: the double brace notation has not been introduced and is quite important for this paper.

12. Line 174: I understand that, when you say "graph" here, you mean a graph together with a vertex coloring function, as well as a filtration defined on those colors.
However, in line 198 (Theorem 1), the graphs $G$ and $G'$ don't come with a filtration.
The convention on what exactly is a graph and what extra structure is used in each result should be made more explicit.
If graphs are always assumed to be colored, I would also suggest using the term "colored graph", or a term to the same effect.

13. Line 198: In Theorem 1, when you write $D_G$, do you mean diagrams in homological degree 0, in homological degree 1, or both?

14. Line 219: Could you please comment on the relevance of Lemma 7? How can it be used (in theory or in practice)?

15. Line 312: Is there a geometric or topological motivation for "case a=0" in line 312?

**Limitations:**

The paper would benefit from a discussion about limitations of the approach, even if brief. This will help readers asses how useful is their approach for specific tasks, as well as what remains to be done and what are the current promising avenues in the theoretical front.

---

> ### Author Rebuttal · Authors · 2023-08-09
>
> We are grateful for your insightful comments and suggestions to improve the paper. Below we address your concerns.
>
> **W1/Q1: "the empirical claims about the performance of the RePHINE would be better substantiated with experimental evaluation on more datasets." / "How does your main strategy (RePHINE) compare to ... Perslay"**
>
> Our initial experiments aimed to 1) corroborate our analysis (synthetic datasets), 2) show that RePHINE also works in scenarios where topological descriptors are combined with GNNs for tackling real-world problems. In total, we considered eight datasets. To further assess the effectiveness of RePHINE, we are currently running experiments regarding integrating RePHINE into SOTA GNNs, including results on additional datasets. We will report these additional experiments in the revised paper.
>
> Based on your feedback, we have now also conducted experiments to compare PersLay and RePHINE on 4 real datasets. To ensure a fair comparison, we processed extended and RePHINE diagrams similarly. In particular, we followed the design of PersLay, i.e., the vectorizations of the diagrams are combined with graph-level features (same as the one used by PersLay) and treated with a linear classifier. We ensured that both methods use identical data samples and used early stopping with the same patience for both methods. The results are in Table 1 of the attached PDF. Overall, RePHINE outperforms PersLay by a significant margin.
>
> **W2: "the limitations of the approach are not discussed in detail...no future work or open questions are mentioned."**
>
> While the complete characterization of the expressivity of RePHINE is an interesting open problem, we can lower-bound its capacity. In particular, if two graphs have one color, RePHINE cannot separate graphs of equal size with the same number of components and cycles. For example, we now show that RePHINE can't separate 4-node star/path graphs (details in the attached PDF).
>
> Importantly, there are many relevant open questions regarding RePHINE/PH in graph learning, including generalization capabilities of existing methods, local versions of RePHINE, and the characterization of which graph properties RePHINE (and other PH-based methods) can compute. Comparing RePHINE and extended PH (or assessing the power of an extended variant of RePHINE) from a theoretical perspective is another interesting open problem. We will add this discussion to the subsection 'Limitations/Future Works' in the revised version of the paper.
>
> **Q2: "Is it clear that RePHINE is isomorphism invariant?"**
>
> Thank you for raising this question that has led us to formally prove that RePHINE is indeed isomorphism invariant as a new Corollary:
>
> *Let $G$ and $G'$ be isomorphic graphs. Then, any edge-color and vertex-color filtrations produce identical RePHINE diagrams for $G$ and $G'$.*
>
> We sketch the essential arguments of the proof here. RePHINE diagram’s isomorphism invariance stems from the fact that it is a function of a filtration on a graph, and this filtration is obtained from isomorphism invariant colorings. Further, when matching a vertex with a diagram element (i.e. deciding which vertex 'died' at the death of a component), RePHINE uses minimum and maximum -functions, which are invariant with respect to the order of comparisons done.
>
> **Q3: "Is RePHINE stable in the sense of persistent homology?"**
>
> Thank you for the interesting question. We believe analyzing the stability properties of RePHINE could be an interesting follow-up work. We will mention this in the newly added subsection ‘Limitations and Future Works’.
>
> **Q4: "you mention that you aim to fully characterize graphs that can be distinguished with persistent homology. Was this achieved or are there still open questions?"**
>
> We analyzed the general case of filtrations based on node and edge colors and indeed provided a complete characterization of attributed graphs that can be distinguished with PH methods that employ these filtrations (using the new notion of color-separating sets). However, there are other types of filtration functions, e.g., based on the spectral decomposition of graph Laplacians, that we have not considered in this paper. Also, there are important open problems, including generalization, stability, and complete characterization of the proposed method, i.e., RePHINE. We believe the novel analyses introduced as part of this work could help in resolving these open problems, and may also foster the rise of other powerful topological descriptors.
>
> Below, we address the minor issues you pointed out.
> 1. **Should the subset inclusion be $\in$**: Yes.
> 2. **the double brace notation has not been introduced**: We introduced the notation for multisets in the revised paper.
> 3. **when you say "graph" here, you mean a graph together with a vertex coloring function, as well as a filtration defined on those colors ...**: We note that filtration functions do not come with our definition of graphs. We have added the term 'colored (or attributed) graphs' when we define graphs.
> 4. **In Theorem 1..do you mean diagrams in homological degree 0, in homological degree 1, or both?** We meant 0-dim diagrams. We have replaced $\mathcal{D}_G$ with $\mathcal{D}_G^0$ for clarity.
> 5. **Could you please comment on the relevance of Lemma 7? How can it be used (in theory or in practice)?**
> Lemmas 6 and 7 motivate the introduction of color-disconnecting sets and help to characterize the expressivity of almost holes. We have added a reference also to Lemma 7 when introducing color-disconnecting sets.
> 6. **Is there a geometric or topological motivation for "case a=0"?** Case a = 0 corresponds to pairs that are augmented with an independent vertex-color filtration.
> ---
> We hope our answers (including empirical comparison with PersLay, proof of isomorphism invariance, and discussion about open problems and limitations) have sufficiently addressed most of your concerns and that you would kindly consider increasing your score.

---

> > ### Comment · Reviewer_x2he · 2023-08-16
> > **Answer to rebuttal**
> >
> > I thank the authors for their response.
> >
> > Overall, I am satisfied with their answers, and I have raised my score accordingly. My score is not higher mainly due to some unanswered theoretical questions (stability and characterization of the expressivity of RePHINE, and comparison to expressivity of extended persistence).

---

### Official Review · Reviewer_PDW5 · 2023-07-08

**Soundness:** 4 excellent
**Presentation:** 3 good
**Contribution:** 3 good
**Rating:** 7
**Confidence:** 4

**Summary:**

The paper provides a theoretical analysis of the expressive power of Persistent Homology (PH) features in distinguishing different colored graphs. The paper characterizes the family of graphs that is separable by a 0-dimensional PH using either node filtering or edge filtering and identifies the failure cases. Based on the theoretical analysis, a new PH filtration is proposed that overcomes the previous limitation and is provably strictly more expressive than either node or edge filtering. The new filtering is compared with standard PH filtering on a synthetic dataset and a few graph-classification benchmarks.

**Strengths:**

The main contribution of the paper lies in the theoretical analysis of the expressive power of PH filtering, which allows one to understand the limits of standard filtering schemes fully and to build a new enriched scheme to overcome them.
The paper is generally well-written and drives you through the reasoning that led to the development of the method while introducing all the essential theorems. Fully understanding them and grasping all the implications requires some effort from the reader, and possible to jump forth and backward from the main paper and the supplementary where proofs are given, but this has to be expected in this kind of paper.

**Weaknesses:**

If the paper is well structured for what concerns the theoretical analysis, the experimental one is a bit lacking. In particular, in its current state, the paper does not position itself with respect to recent methods for graph classification, and it is not clear if it is a practical competitive alternative or if the contribution mostly lies on the theoretical analysis and just paves the road to future possible developments.

A felt also that an analysis of the expressive power bounds of the proposed method could have been interesting. For instance, is there any particular graph structure in which the proposed method cannot distinguish two non-isometric graphs?

**Questions:**

Going through the Lemmas introduced in the preliminaries and the following section, it is not clear which lemmas/theorems are introduced by the authors and which ones are just reported (if any).

I would add to the comparison soma SOTA method for graph classification.

For MOLHIV the ROC-AUC is usually reported.

**Limitations:**

I do not foresee any particular negative societal impact. A discussion on the theoretical and practical (if any) limitations of the proposed method, also w.r.t. SOTA on graph classification, would better help to understand the current and future potential of the method.

---

> ### Author Rebuttal · Authors · 2023-08-09
>
> Thank you very much for your thoughtful feedback. We address all your questions below.
>
> **W1: "the paper does not position itself with respect to recent methods for graph classification"**
>
> Thanks for the opportunity to position our work appropriately.  Persistent homology methods that we consider here provide valuable topological information that can often be integrated into recent graph embedding methods such as variants of (geometric) graph neural networks (GNNs) to boost their performance. Therefore, we believe their role is complementary to the existing graph classification methods, so our initial experiments focused on corroborating the benefits of augmenting popular GNNs with topological descriptors. We are currently running experiments to investigate further benefits of integrating the RePHINE diagrams into modern GNNs (e.g., PNA, SpecFormer). We will include these results in the revised version of the manuscript.
>
> Based on the reviews, we have now also compared the proposed method RePHINE with a state-of-the-art persistent homology method Extended PH (Perslay) on several real datasets such as NCI109, Proteins, IMDB-B, and NCI1. Our results demonstrate that RePHINE performs better on these datasets (please see Table 1 in the attached pdf). Thus, beyond the theoretical analyses, the empirical benefits of RePHINE can already be demonstrated.
>
> **W2: "is there any particular graph structure in which the proposed method cannot distinguish two non-isometric graphs?"**
>
> That's another excellent question. Indeed, there are non-isomorphic graphs that cannot be separated based on RePHINE diagrams. In particular, if two graphs have one color, RePHINE cannot separate graphs of equal size with the same number of components and cycles. For instance, a 4-node star graph and a 4-node path graph cannot be separated. We've added a visualization in Figure 1 (in the attached pdf) to show this limitation.
>
> **Q1: "it is not clear which lemmas/theorems are introduced by the authors and which ones are just reported."**
>
> All Lemmas/Theorems,  including those in the preliminaries and following sections, are introduced and proven in this paper. To emphasize this, we've now added a Table with a summary of our contributions in the Introduction --- we've also included the Table in the attached PDF. Please note that we have now additionally proved that RePHINE is isomorphism invariant (see our reply to reviewer x2he for details).
>
> **Q2: "I would add to the comparison soma SOTA method for graph classification."**
>
> Thanks for your comment. As we mentioned in the answer to R1-W1 above, we are running further experiments to substantiate the benefits of integrating RePHINE into SOTA GNNs. We will include these results in the revised version.
>
> **Q3: "For MOLHIV the ROC-AUC is usually reported."**
>
> Thanks for catching this. Indeed, the numbers in the paper for MOLHIV are ROC-AUC values. We will make this clear in the revised manuscript.
>
> ***
> Many thanks again for your constructive feedback. Based on your review, we will update our paper to include additional results regarding the combination of RePHINE + modern GNNs (SOTA) and clarify our contributions (including their limits). We hope our answers have sufficiently addressed your concerns, and the same translates into your stronger support for this work.

---

> > ### Comment · Reviewer_PDW5 · 2023-08-14
> > **Thanks for the clarification**
> >
> > Dear Authors,
> > thanks for taking the time to answer my doubts. The newly added material and experiments seem convincing. Should you have some preliminary results on the use of your method with SOTA GNNs before the reviewers/authors discussion period, I would be curious to see them.

---

> > > ### Author Response · Authors · 2023-08-18
> > > **Results using PNA on ZINC**
> > >
> > > Thanks for your feedback. We are glad to hear you found the newly added material and experiments convincing.
> > >
> > > Regarding experiments with SOTA GNNs, we run a fair comparison between PNA and PNA+RePHINE on the ZINC dataset (public splits). We leverage the topological descriptors as described in the paper (see equations in Section 4). Both methods use the same hyper-parameters (available at the PyTorch-geometric toolbox) and training procedures. **The results obtained over ten independent runs with different seeds are: PNA ($0.195 \pm 0.004$ MAE) and PNA+RePHINE ($0.189 \pm 0.006$ MAE)**. In this case, RePHINE improves the performance of PNA, achieving MAE that lies one standard deviation away from that of PNA alone. Moreover, we plan to consider at least a Transformer-based architecture and another OGB dataset in the final paper.

---

### Author Rebuttal · Authors · 2023-08-09

We are grateful to all the reviewers for their time and insightful comments, as well as to the (senior) area, program, and general chairs for their service to the community.

We are glad to note the positive response of all the reviewers, and specifically, their acknowledgments that our work is **interesting and novel** (x2he, BFX1) and **provides a resolution to the problem of recognizing attributed graphs** based on the persistence of their connected components (oULm). Also, reviewers found that our work can **allow one to understand the limits of standard filtering schemes** and **build new enriched schemes to overcome them** (PDW5). Finally, our **theoretical contributions are said to be well presented and clear** (rXNh, BFX1, PDW5), **relevant in practice** (rXNh, x2he), and **supported by experiments on eight datasets** (rXNh).

To the best of our efforts, we’ve tried to address all the specific comments, including the minor ones, that have been raised by each reviewer. In particular, some of the main revisions are:

- Additional experimental results on the comparison between RePHINE and Extended Persistence Diagrams (PersLay);
- Proof that RePHINE is isomorphism invariant (newly added Corollary 1);
- New subsection about 'Limitations and Future Works';
- Clarifications regarding the main contributions (newly added Table with an overview of our theoretical results), and exposition of RePHINE diagrams (now as a formal definition);
- Added visualizations about the graphs and diagrams obtained from the synthetic experiments.

Moreover, we are currently running experiments to show additional results on 1) the combination of RePHINE with SOTA GNNs; 2) more (larger) datasets.

We believe that acting on reviewers’ feedback has reinforced the many strengths of this work, and we thank them again for their very constructive comments.

---

### Decision · Program_Chairs · 2023-09-21

**Decision:**

Accept (oral)

**Comment:**

This paper and the ensuing discussion was a joy to read and behold. It presents a direly-needed investigation into the theoretical limits of persistent homology in graph learning, while developing new ways to leverage topological information in neural networks. The paper is technically sound and well-written—a fact that was acknowledged by all reviewers—and I fully agree with the reviewers here. It is thus my pleasure to **endorse this paper for presentation at NeurIPS**.

Moving forward, the authors are encouraged to further improve the text by positioning and contextualising their work, with a particular eye towards theoretical or empirical limitations of their work.